# 3D molecule generation by denoising voxel grids

**Pedro O. Pinheiro, Joshua Rackers, Joseph Kleinhenz, Michael Maser, Omar Mahmood, Andrew Martin Watkins, Stephen Ra, Vishnu Sresht, Saeed Saremi**

Prescient Design, Genentech

## Abstract

We propose a new score-based approach to generate 3D molecules represented as atomic densities on regular grids. First, we train a denoising neural network that learns to map from a smooth distribution of noisy molecules to the distribution of real molecules. Then, we follow the *neural empirical Bayes* framework [1] and generate molecules in two steps: (i) sample noisy density grids from a smooth distribution via underdamped Langevin Markov chain Monte Carlo, and (ii) recover the "clean" molecule by denoising the noisy grid with a single step. Our method, *VoxMol*, generates molecules in a fundamentally different way than the current state of the art (*i.e.*, diffusion models applied to atom point clouds). It differs in terms of the data representation, the noise model, the network architecture and the generative modeling algorithm. Our experiments show that VoxMol captures the distribution of drug-like molecules better than state of the art, while being faster to generate samples.

## 1 Introduction

Finding novel molecules with desired properties is an important problem in chemistry with applications to many scientific domains. In drug discovery in particular, standard computational approaches perform some sort of local search—by scoring and ranking molecules—around a region of the molecular space (chosen based on some prior domain knowledge). The space of possible drug-like molecules is prohibitively large (it scales exponentially with the molecular size [2, 3], estimated to be around $10^{60}$ [4]), therefore search in this space is very hard. Search-based approaches achieve some successes in practice, but have some severe limitations: we can only explore very small portions of the molecular space (on the order of billions to trillions molecules) and these approaches cannot propose new molecules conditioned on some desiderata.

Generative models for molecules have been proposed to overcome these limitations and explore the molecular space more efficiently [5]. These approaches often consider one of the following types of molecule representations: (i) one-dimensional sequences such as SMILES [6] or SELFIES [7] (*e.g.*, [8, 9, 10]), (ii) two-dimensional molecular graphs, where nodes represent atoms or molecular substructures and edges represent bonds between them (*e.g.*, [11, 12, 13, 14]), or (iii) atoms as three-dimensional points in space. Molecules are entities laying on three-dimensional space, therefore 3D representations are arguably the most complete ones—they contain information about atom types, their bonds and the molecular conformation.

Recent generative models consider molecules as a set of points in 3D Euclidean space and apply diffusion models on them [15, 16, 17, 18, 19, 20]. Point-cloud representations allow us to use equivariant graph neural networks [21, 22, 23, 24, 25]—known to be very effective in molecular discriminative tasks—as the diffusion model's denoising network. However, point-based diffusion approaches have some limitations when it comes to generative modeling. First, the number of atoms in the molecule (*i.e.*, nodes on the 3D graph) to be diffused need to be known beforehand. Second, atom types and their coordinates have very different distributions (categorical and continuous variables, respectively) and are treated separately. Because a score function is undefined on discrete distributions,

37th Conference on Neural Information Processing Systems (NeurIPS 2023).

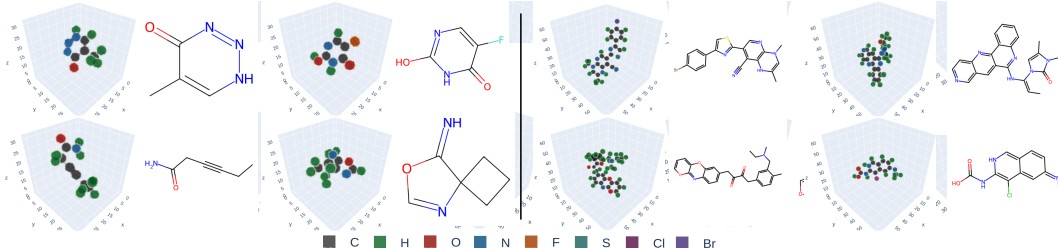

Figure 1: Voxelized molecules generated by our model and their corresponding molecular graphs. Left, samples from a model trained on QM9 dataset ($32^3$ voxels). Right, samples from a model trained on GEOM-drugs ($64^3$ voxels). In both cases, each voxel is a cubic grid with side length of .25Å. Each color represents a different atom (and a different channel on the voxel grid). Best seen in digital version. See appendix for more generated samples.

some workaround is necessary. Finally, graph networks operate only on nodes and edges (single and pairwise iterations, respectively). Therefore, capturing long-range dependencies over multiple atoms (nodes) can become difficult as the number of atoms increases. This is related to the limitations of the message-passing formalism in graph neural networks [26]. Higher-order message passing can alleviate this problem to a degree [27, 28], but they come at a significant computational cost and they have been limited to third-order models [29] (see next section for more discussions on the tradeoffs between model expressivity and built-in equivariance).[1]

In this work we introduce *VoxMol*, a new score-based method to generate 3D molecules. Similar to [33], and unlike most recent approaches, we represent atoms as continuous (Gaussian-like) densities and molecules as a discretization of 3D space on voxel (*i.e.*, a discrete unit of volume) grids. Voxelized representations allow us to use the same type of denoising architectures used in computer vision. These neural networks—the workhorse behind the success of score-based generative models on images, *e.g.* [34, 35, 36]—are very effective and scale very well with data.

We start by training a neural network to denoise noisy voxelized molecules. Noisy samples are created simply by adding Gaussian noise (with a fixed identity covariance matrix scaled by a large noise level) to each voxel in the molecular grid. This denoising network also parametrizes the score function of the smooth/noisy distribution. Note that in contrast to diffusion models, the noise process we use here does not displace atoms. Then, we leverage the (learned) denoising network and generate molecules in two steps [1]: (i) *(walk)* sample noisy density grids from the smooth distribution via Langevin Markov chain Monte Carlo (MCMC), and (ii) *(jump)* recover "clean" molecules by denoising the noisy grid. This sampling scheme, referred to as walk-jump sampling in [1], has been successfully applied before to 2D natural images [37, 38] and 1D amino acid sequences [39].

Compared to point-cloud diffusion models, VoxMol is simpler to train, it does not require knowing the number of atoms beforehand, and it does not treat features as different distributions (continuous, categorical and ordinal for coordinates, atom types and formal charge)—we only use the "raw" voxelized molecule. Moreover, due to its expressive network architecture, our method scales better to large, drug-sized molecules. Figure 1 (and Figures 8, 9 on appendix) illustrates voxelized molecules and their corresponding molecular graphs generated by our model, trained on two different datasets. These samples show visually that our model learns valences of atoms and symmetries of molecules.

The main contributions of this work can be summarized as follows. We present VoxMol, a new score-based method for 3D molecule generation. The proposed method differs from current approaches—usually diffusion models on point clouds—in terms of the data representation, the noise model, the network architecture, and the generative modeling algorithm. We show in experiments that VoxMol performs slightly worse than state of the art on a small dataset (QM9 [40]), while outperforms it (by a large margin) on a challenging, more realistic drug-like molecules dataset (GEOM-drugs [41]).

---

[1]On this topic and since our approach in 3D molecule generation is based in computer vision, we should also highlight here the great success of vision transformer models [30], where in contrast to convolutional neural networks [31] no notion of equivariance whatsoever is built into the model architecture. Quite remarkably, after training, vision transformers can achieve a *higher* degree of equivariance compared to convolutional architectures [32].

## 2   Related Work

**Voxel-based unconditional 3D molecule generation.**    Skalic *et al.* [42] and Ragoza *et al.* [33] map atomic densities on 3D regular grids and train VAEs [43] using 3D convolutional networks to generated voxelized molecules. To recover atomic coordinates from the generated voxel grids[2], [33] introduces a simple optimization-based solution, while [42] trains another model that "translates" voxel structures into SMILES strings. Voxel representations are flexible and can trivially be applied to related problems with different data modalities. For instance, [44] proposes a GAN [45] on voxelized electron densities, while [46] leverages voxelized 3D pharmacophore features to train a pocket-conditional model. Similar to these works, our model also relies on discretization of 3D space. Like [33], we use a simple peak detection algorithm to extract atomic coordinates from the generated voxel grids. However, our method differs on the underlying generative modeling, architecture, datasets, input representations and evaluations.

**Point cloud-based unconditional generation.**    Most recent models treat molecules as sets of points, where each node is associated with a particular atom type, its coordinates and potentially extra information like formal charge. Different modeling approaches have been proposed, *e.g.*, [47, 48, 49] utilize autoregressive models to iteratively sample atoms, and [50, 51] use normalizing flows [52]. Hoogeboom *et al.* [15] proposes E(3) Equivariant Diffusion Models (EDM), a diffusion [53]-based approach that performs considerably better than previous models on this task. EDMs learn to denoise a diffusion process (operating on both continuous and categorical data) and generate molecules by iteratively applying the denoising network on an initial noise. Several works have been proposed on the top of EDM [54, 55, 20, 56]. For instance, Xu *et al.* [56] improves EDM by applying diffusion on a latent space instead of the atomic coordinates, while MiDi [20] shows that EDM results can be improved by jointly generating the 3D conformation and the connectivity graph of molecules (in this setting, the model has access to both the 3D structure and the 2D connectivity graph).

**Conditional 3D molecule generation.**    A related body of work is concerned with *conditional* generation. In many cases, conditional generation is built on the top of unconditional generation methods. Some authors propose to predict the 3D structure of the molecules given a molecular graph (this is called the conformer generation task): VAEs [57, 58], normalizing flows [59], reinforcement learning [60], optimal transport [61], autoregressive models [62] and diffusion models [63, 16, 64] have been proposed to this task. Some work [65, 66] condition 3D generation on shape while other works condition molecule generation on other structures. For instance, [17, 18, 19, 67] adapt (unconditional) diffusion models to condition on protein pockets, while [68] adapt their previous work [33] to condition voxelized structures to protein targets. Finally, [46] proposes a hybrid conditional generation model by modeling fragments/scaffolds with point cloud representation and the 3D target structures and pharmacophores features [69] with voxel grids.

**Comparison between voxel and point-cloud representations.**    Voxels have some advantages and disadvantages compared to point cloud representations. First, voxels are straightforward generalizations of 2D pixels to 3D space, therefore we can leverage similar machinery used in score-based generative modeling for images. These models are known to perform well and scale nicely with data. Second, message passing on graphs operate on single and pairwise interactions while convolution filters (and potentially transformer layers applied to regular grids) can capture multiple local interactions by construction (see [70] for a discussion on the *many-body representation* hypothesis). Third, voxel representations have a higher memory footprint and lower random memory accesses than point cloud representations [71]. We note however, that developing models on drug-sized molecules (that is, molecules with size close to those on GEOM-drugs [41]) with reasonable resolution (.1–.2Å) is possible on current GPU hardware. Fourth, recovering point coordinates from a discrete grid has no analytical solution, therefore voxel-based models require an extra step to retrieve atomic coordinates. We show empirically that this is not a problem in practice as we can achieve competitive results, even with a very simple peak detection algorithm.

Finally, graph networks are less expressive due to message passing formalism [26, 27], but are a better fit for built-in SE(3)-equivariance architectures (*e.g.* [21, 22, 23, 24, 25]). Rotation-equivariant 3D con-

---

[2]This is necessary if we want to extract the molecular graph. However, raw voxelized generations could be useful to other computational chemistry tasks.

volutional network have been proposed [72, 73, 74] [3], but current models do not scale as well as standard convnets, and it would be a challenge to apply them to drug-sized molecules. Built-in rotation equivariance is a good property to have, however equivariance can also be learned with strong data augmentation/larger datasets [75, 76, 32]. In fact, concurrently to this work, [77] also show that built-in SE(3)-equivariant architecture is not necessary to generate molecules. Our experiments show that an expressive denoiser scales up better, allowing VoxMol to outperform current state of the art on GEOM-drugs. However, we hope our results motivate exploration of more efficient SE(3)-equivariant convnet architectures.

## 3    Method

We follow previous work (*e.g.*, [78, 33, 70, 79]) and represent atoms as continuous Gaussian-like atomic densities in 3D space, centered around their atomic coordinates. Molecules are generated by discretizing the 3D space around the atoms into voxel grids, where each atom type (element) is represented by a different grid channel. See appendix for more information on how we discretize molecules. This discretization process gives us a dataset with $n$ *voxelized molecules* $\{x_i\}_{i=1}^n, x_i \in \mathbb{R}^d, d = c \times l^3$, where $l$ is the length of each grid edge and $c$ is the number of atom channels in the dataset. Each voxel in the grid can take values between 0 (far from all atoms) and 1 (at the center of atoms). Throughout our experiments, we consider a fixed resolution of .25Å (we found it to be a good trade-off between accuracy and computation). Therefore, voxel grids occupy a volume of $(l/4)^3$ cubic Ångströms.

### 3.1    Background: neural empirical Bayes

Let $p(x)$ be an unknown distribution of voxelized molecules and $p(y)$ a smoother version of it obtained by convolving $p(x)$ with an isotropic Gaussian kernel with a known covariance $\sigma^2 I_d$ [4]. Equivalently, $Y = X + N$, where $X \sim p(x)$, $N \sim \mathcal{N}(0, \sigma^2 I_d)$. Therefore $Y$ is sampled from:

$$p(y) = \int_{\mathbb{R}^d} \frac{1}{(2\pi\sigma^2)^{d/2}} \exp\left(-\frac{\|y - x\|^2}{2\sigma^2}\right) p(x) dx.$$

This transformation will smooth the density of $X$ while still preserving some of the structure information of the original voxel signals. Robbins [80] showed that if we observe $Y = y$, then the least-square estimator of $X$ *is* the Bayes estimator, *i.e.*, $\hat{x}(y) = \mathbb{E}[X|Y = y]$. Built on this result, Miyasawa [81] showed that, if the noising process is Gaussian (as in our case), then the least-square estimator $\hat{x}(y)$ can be obtained purely from the (unnormalized) smoothed density $p(y)$:

$$\hat{x}(y) = y + \sigma^2 g(y), \tag{1}$$

where $g(y) = \nabla_y \log p(y)$ is the score function [82] of $p(y)$. This interesting equation tells us that, *if* we know $p(y)$ up to a normalizing constant (and therefore the score function associated with it), we can estimate the original signal $x$ only by observing its noisy version $y$. Equivalently, if we have access to the estimator $\hat{x}(y)$, we can compute the score function of $p(y)$ via (1).

Our generative model is based on the *neural empirical Bayes (NEB)* formalism [1]: we are interested in learning the score function of the smoothed density $p(y)$ and the least-square estimator $\hat{x}(y)$ from a dataset of voxelized molecules $\{x_i\}_{i=1}^n$, sampled from unknown $p(x)$. We leverage the (learned) estimator and score function to generate voxelized molecules in two steps: (i) sample $y_k \sim p(y)$ with Langevin MCMC [83], and (ii) generate clean samples with the least-square estimator. The intuition is that it is much easier to sample from the smooth density than the original distribution. See Saremi and Hyvärinen [1] for more details.

### 3.2    Denoising voxelized molecules

We parametrize the Bayes estimator of $X$ using a neural network with parameters $\theta$ denoted by $\hat{x}_\theta : \mathbb{R}^d \to \mathbb{R}^d$. Since the Bayes estimator is the least-squares estimator, the learning becomes a least-squares *denoising objective* as follows:

$$\mathcal{L}(\theta) = \mathbb{E}_{x \sim p(x), y \sim \mathcal{N}(x, \sigma^2 I_d)} \|x - \hat{x}_\theta(y)\|^2. \tag{2}$$

---

[3]We made an attempt at using an equivariant 3D convnets for denoising, but initial experiments were not successful. See appendix for details.

[4]We use the convention where we drop random variable subscripts from probability density function when the arguments are present: $p(x) := p_X(x)$ and $p(y) := p_Y(y)$.

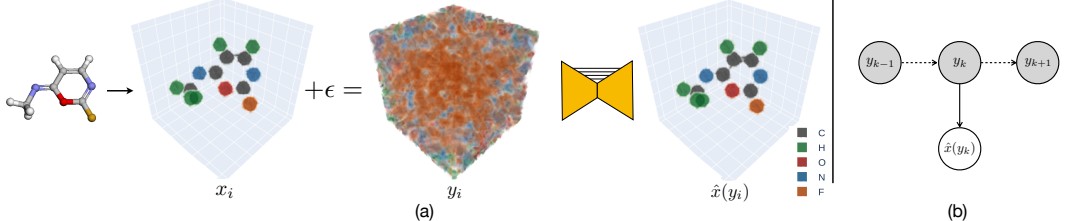

Figure 2: (a) A representation of our denoising training procedure. Each training sample (*i.e.*, a voxelized molecule) is corrupted with isotropic Gaussian noise with a fixed noise level $\sigma$. The model is trained to recover clean voxel grids from the noisy version. To facilitate visualization, we threshold the grid values, $\hat{x} = \mathbb{1}_{\geq .1}(\hat{x})$. (b) Graphical model representation of the walk-jump sampling scheme. The dashed arrows represent the walk, a MCMC chain to draw noisy samples from $p(y)$. The solid arrow represents the jump. Both walks and jumps leverage the trained denoising network.

Using (1), we have the following expression for the smoothed score function in terms of the denoising network[5]:

$$g_\theta(y) = \frac{1}{\sigma^2}(\hat{x}_\theta(y) - y). \tag{3}$$

By minimizing the learning objective (2) we learn the optimal $\hat{x}_\theta$ and by using (3) we can compute the score function $g_\theta(y) \approx \nabla_y \log p(y)$.

We model the denoising network $\hat{x}_\theta$ with an encoder-decoder 3D convolutional network that maps every noised voxel on the grid to a clean version of it. Figure 2(a) shows a general overview of the denoising model. The noise level, $\sigma$, is kept constant during training and is a key hyperparameter of the model. Note that in the empirical Bayes formalism, $\sigma$ can be any (large) value.

Compared to diffusion models, this training scheme is simpler as the noise level is fixed during training. VoxMol does not require noise scheduling nor temporal embedding on the network layers. We observe empirically that single-step denoising is sufficient to reconstruct voxelized molecules (within the noise levels considered in this paper). Our hypothesis is that this is due to the nature of the voxel signals, which contain much more "structure" than "texture" information, in comparison to natural images.

### 3.3 Sampling voxelized molecules

We use the learned score function $g_\theta$ and the estimator $\hat{x}_\theta$ to sample. We follow the *walk-jump sampling* scheme [1, 37, 38, 39] to generate voxelized molecules $x_k$:

(i) *(walk step)* For sampling noisy voxels from $p(y)$, we consider Langevin MCMC algorithms that are based on discretizing the underdamped Langevin diffusion [84]:

$$\begin{aligned} dv_t &= -\gamma v_t dt - u g_\theta(y_t) dt + (\sqrt{2\gamma u}) dB_t \\ dy_t &= v_t dt, \end{aligned} \tag{4}$$

where $B_t$ is the standard Brownian motion in $\mathbb{R}^d$, $\gamma$ and $u$ are hyperparameters to tune (friction and inverse mass, respectively). We use the discretization algorithm proposed by Sachs *et al.* [85] to generate samples $y_k$, which requires a discretization step $\delta$. See appendix for a description of the algorithm.

(ii) *(jump step)* At an arbitrary time step $k$, clean samples can be generated by estimating $X$ from $y_k$ with the denoising network, *i.e.*, computing $x_k = \hat{x}_\theta(y_k)$.

This approach allows us to approximately sample molecules from $p(x)$ without the need to compute (or approximate) $\nabla_x \log p(x)$. In fact, we do MCMC on the smooth density $p(y)$, which is known to be easier to sample and mixes faster than the original density $p(x)$ [1, 38, 86]. Figure 2(b) shows a schematic representation of the generation process. Following [37], we initialize the chains at by adding uniform noise to Gaussian noise (with the same $\sigma$ used during training), *i.e.*, $y_0 = N + U$, $N \sim \mathcal{N}(0, \sigma^2 I_d)$, $U \sim \mathcal{U}_d(0,1)$ (this was observed to mix faster in practice).

---

[5]Alternatively, one can also parameterize the score function: $g_\theta : \mathbb{R}^d \to \mathbb{R}^d$, in which case $\hat{x}_\theta(y) = y + \sigma^2 g_\theta(y)$.

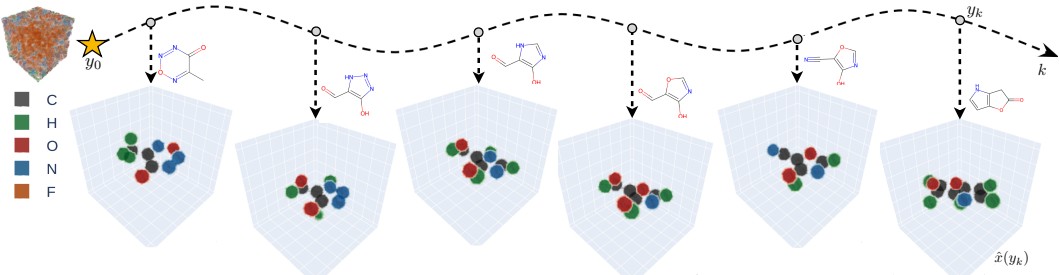

Figure 3: Illustration of walk-jump sampling chain. We do Langevin MCMC on the noisy distribution (walk) and estimate clean samples with the denoising network at arbitrary time (jump).

The noise level plays a key role in this sampling framework. If the noise is low, denoising (jump step) becomes easier, with lower variance, while sampling a "less smooth" $p(y)$ (walk step) becomes harder. If the noise is high, the opposite is true.

Figure 3 illustrates an example of a walk-jump sampling chain, where generated molecules change gradually as we walk through the chain (the clean samples are shown every ten steps, $\Delta k = 10$). This figure is a demonstration of the fast-mixing properties of our sampling scheme in generating 3D molecules. For instance, some atoms (or other structures like rings) might appear/disappear/change as we move through the chain. Interestingly, this behavior happened on most chains we looked into explicitly.

### 3.4 Recovering atomic coordinates from voxelized molecules

It is often useful to extract atomic coordinates from generated voxelized molecules (*e.g.*, to validate atomic valences and bond types or compare with other models). We use a very simple algorithm (a simplified version of the approach used in [33]) to recover the set of atomic coordinates from generated voxel grids: first we set to 0 all voxels with value less than .1, *i.e.*, $x_k = \mathbb{1}_{\geq .1}(x_k)$. Then we run a simple peak detection to locate the voxel on the center of each Gaussian blob (corresponding to the center of each atom). Finally we run a simple gradient descent coordinate optimization algorithm to find the set of points that best create the generated voxelized molecule. Once we have obtained the optimized atomic coordinates, we follow previous work [33, 18, 17, 20] and use standard cheminformatics software to determine the molecule's atomic bonds. Figure 4 shows our pipeline to recover atomic coordinates and molecular graphs from generated voxelized molecules. See appendix for more details.

## 4 Experiments

In this section, we evaluate the performance of our model on the task of unconditional 3D molecule generation. Our approach is the first of its kind and therefore the objective of our experiments is to show that (i) VoxMol is a feasible approach for unconditional generation (this is non-trivial) and (ii) it scales well with data, beating a established model on a large, drug-like dataset. In principle, VoxMol can be used for guided (or conditional) generation, an arguably more useful application for molecular sciences (see appendix for a discussion on how guidance can be used on generation).

We start with a description of our experimental setup, followed by results on two popular datasets for this problem. We then show ablation studies performed on different components of the model.

### 4.1 Experimental setup

**Architecture.** The denoising network $\hat{x}_\theta$ is used in both the walk and jump steps described above. Therefore, its parametrization is very important to the performance of this approach. We use a 3D U-Net [87] architecture for our denoising network. We follow the same architecture recipe as DDPM [34], with two differences: we use 3D convnets instead of 2D and we use fewer channels on all layers. The model has 4 levels of resolution and we use self-attention on the two lowest resolutions. We augment our dataset during training by applying random rotation and translation to every training sample. Our models are trained with noise level $\sigma = .9$, unless stated otherwise. We train our models with batch size of 128 and 64 (for QM9 and GEOM-drugs, respectively) and we use AdamW [88]

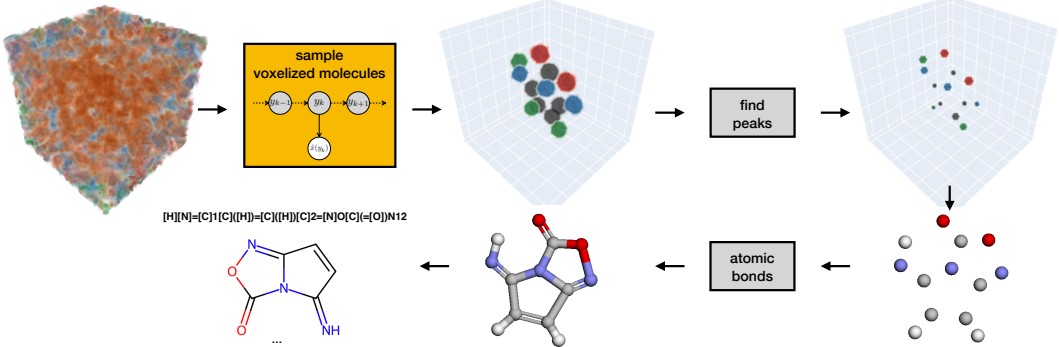

Figure 4: Pipeline for recovering atomic coordinates from voxel grids: (i) VoxMol generates voxelized molecules, (ii) atomic coordinates are extracted from voxel grid with simple peak detection algorithm, (iii) we use cheminformatics software to add atomic bonds and extract SMILES strings, molecular graphs, etc.

(learning rate $2 \times 10^{-5}$, weight decay $10^{-2}$) to optimize the weights. The weights are updated with exponential moving average with a decay of .999. We use $\gamma = 1.0$, $u = 1.0$ and $\delta = .5$ for all our MCMC samplings. See appendix for more details on the architecture, training and sampling.

**Datasets.** We consider two popular datasets for this task: *QM9* [40] and *GEOM-drugs* [41]. QM9 contains small molecules with up to 9 heavy atoms (29 if we consider hydrogen atoms). GEOM-drugs contains multiple conformations for 430k drug-sized molecules and its molecules have 44 atoms on average (up to 181 atoms and over 99% are under 80 atoms). We use grids of dimension $32^3$ and $64^3$ for QM9 and GEOM-drugs respectively. These volumes are able to cover over 99.8% of all points on both datasets. All our models model hydrogens explicitly. For QM9, we consider all 5 chemical elements (C, H, O, N and F) present on the dataset. For GEOM-drugs, we consider 8 elements (C, H, O, N, F, S, Cl and Br). We ignore P, I and B elements as they appear in less than .1% of the molecules in the dataset. Finally, the input voxel grids are of dimension $\mathbb{R}^{5 \times 32 \times 32 \times 32}$ and $\mathbb{R}^{8 \times 64 \times 64 \times 64}$ for QM9 and GEOM-drugs, respectively. We perform the same pre-processing and dataset split as [20] and end up with 100K/20K/13K molecules for QM9 and 1.1M/146K/146K for GEOM-drugs (train, validation, test splits respectively).

**Baselines.** We compare our method with two state-of-the-art approaches: *GSchNet* [47], a point-cloud autoregressive model and *EDM* [15], a point-cloud diffusion-based model. We note that both methods rely on equivariant networks, while ours does not. Our results could potentially be improved by successfully exploiting equivariant 3D convolutional networks. We also show results of VoxMol$_{\text{oracle}}$ in our main results, where we assume we have access to real samples from the noisy distribution. Instead of performing MCMC to sample $y_k$, we sample molecules from the validation set and add noise to them. This baseline assumes we would have perfect sampling of noisy samples (walk step) and let us assess the quality of our model to recover clean samples. It serves as an upper bound for our model and allows us to disentangle the quality of the walk (sampling noisy samples) and jump (estimating clean molecules) steps.

All methods generate molecules as a set of atom types and their coordinates (in the case of voxelized molecules, we use the post-processing described above to get the atomic coordinates). We follow previous work [33, 18, 17, 20] and use standard cheminformatics software to determine the molecule's atomic bonds given the atomic coordinates [6]. Using the same post-processing for all methods allows a more apples-to-apples comparison of the models.

**Metrics.** Most metrics we use to benchmark our model come from [20] [7]. We draw 10,000 samples from each method and measure performance with the following metrics: *stable mol* and *stable atom*, the percentage of stable molecules and atoms, respectively, as defined in [15]; *validity*, the percentage of generated molecules that passes RDKit [90]'s sanitization filter; *uniqueness*, the proportion of valid molecules that have different canonical SMILES; *valency* $W_1$, the Wasserstein distance between the distribution of valencies in the generated and test set; *atoms TV* and *bonds TV*, the total variation

---

[6]We follow previous work and use OpenBabel [89] to get the bond orders.
[7]We use the official implementation from https://github.com/cvignac/MiDi.

| | stable mol %↑ | stable atom %↑ | valid %↑ | unique %↑ | valency $W_1$↓ | atom TV↓ | bond TV↓ | bond len $W_1$↓ | bond ang $W_1$↓ |
|---|---|---|---|---|---|---|---|---|---|
| *data* | 98.7 | 99.8 | 98.9 | 99.9 | .001 | .003 | .000 | .000 | .120 |
| GSchNet | 92.0 | 98.7 | 98.1 | 94.5 | .049 | .042 | .041 | .005 | 1.68 |
| EDM | 97.9 | 99.8 | 99.0 | 98.5 | .011 | .021 | .002 | .001 | 0.44 |
| VoxMol$_{\text{no rot}}$ | 84.2 (±1.6) | 98.2 (±.3) | 98.1 (±.4) | 77.2 (±1.7) | .043 (±.0) | .171 (±.200) | .050 (±.010) | .007 (±.0) | 3.80 (±.7) |
| VoxMol | 89.3 (±.6) | 99.2 (±.1) | 98.7 (±.1) | 92.1 (±.3) | .023 (±.002) | .029 (±.009) | .009 (±.002) | .003 (±.002) | 1.96 (±.04) |
| VoxMol$_{\text{oracle}}$ | 90.1 | 99.3 | 98.9 | 99.9 | .024 | .009 | .002 | .001 | 0.37 |

Table 1: Results on QM9. We use 10,000 samples from each method. Our results are shown with mean/standard deviation across 3 runs.

between the distribution of atom types and bond types, respectively; *bond length $W_1$* and *bond angle $W_1$*, the Wasserstein distance between the distribution of bond and lengths, respectively. Finally, we also report the *strain energy* metric proposed in [91]. This metric is defined as the difference between the internal energy of the generated molecule's pose and a relaxed pose of the molecule. The relaxation and the energy are computed using the Universal Force Field (UFF) [92] within RDKit. See appendix for more details about the metrics.

## 4.2 Experimental results

Table 1 and Table 2 show results on QM9 and GEOM-drugs respectively. We report results for models trained with and without data augmentation (VoxMol and VoxMol$_{\text{no rot}}$, respectively) and generate 10,000 samples with multiple MCMC chains. Each chain is initialized with 1,000 warm-up steps, as we observed empirically that it slightly improves the quality of generated samples. Then, samples are generated after each 500 walk steps (each chain having a maximum of 1,000 steps after the warm-up steps). Results for our models are shown with mean/standard deviation among three runs. The row *data* on both tables are randomly sampled molecules from the training set.

On QM9, VoxMol performs similar to EDM in some metrics while performing worse in others (specially stable molecule, uniqueness and angle lengths). On GEOM-drugs, a more challenging and realistic drug-like dataset, the results are very different: *VoxMol outperforms EDM in eight out of nine metrics, often by a considerably large margin*.

Figure 5(a,b) shows the cumulative distribution function (CDF) of strain energies for the generated molecules of different models on QM9 and GEOM-drugs, respectively. The closer the CDF of generated molecules from a model is to that of *data* (samples from training set), the lower is the strain energy of generated molecules. The ground truth data has median strain energy of 43.87 and 54.95 kcal/mol for QM9 and GEOM-drugs, respectively. On QM9, all models have median strain energy around the same ballpark: 52.58, 66.32 and 56.54 kcal/mol for EDM, VoxMol$_{\text{no rot}}$ and VoxMol, respectively. On GEOM-drugs, the molecules generated by *VoxMol have considerably lower median strain energy than EDM*: 951.23 kcal/mol for EDM versus 286.06 and 171.57 for VoxMol$_{\text{no rot}}$ and VoxMol.

We observe, as expected, that augmenting the training data with random rotations and translations improves the performance of the model. The improvement is bigger on QM9 (smaller dataset) than on GEOM-drugs. In particular, the augmentations help to capture the distribution of bonds and angles between atoms and to generate more unique molecules. We note that, unlike EDM, our model does not require knowledge of the number of atoms beforehand (neither for training nor sampling). In fact, Figure 6 show that our model learns the approximate distribution of the number of atoms per molecule on both datasets. Implicitly learning this distribution can be particularly useful in applications related to in-painting (*e.g.*, pocket conditioning, linking, scaffold conditioning). Finally, our method generates drug-like molecules in fewer iterations and is faster than EDM on average (see Table 3). EDM sampling time scales quadratically with the number of atoms, while ours has constant time (but scales cubically with grid dimensions).

These results clearly show one of the main advantages of our approach: a more expressive model *scales better* with data. Architecture inductive biases (such as built-in SE(3) equivariance) are helpful in the setting of small dataset and small molecules. However, on the large-scale regime, a more expressive model is more advantageous in capturing the modes of the distribution we want to model. Compared

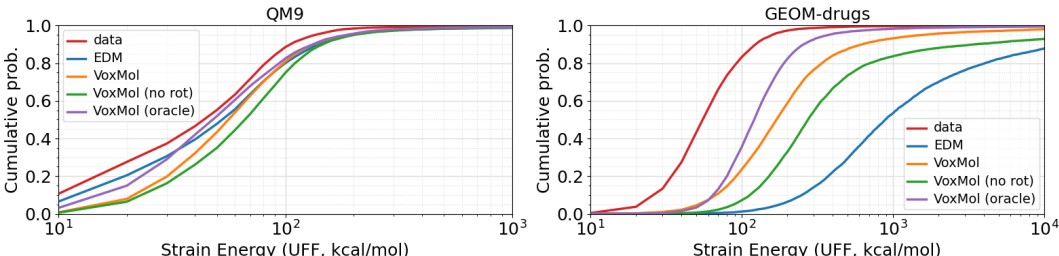

Figure 5: The cumulative distribution function of strain energy of generated molecules on (a) QM9 and (b) GEOM-drugs. For each method, we use 10,000 molecules.

| | stable mol %↑ | stable atom %↑ | valid %↑ | unique %↑ | valency $W_{1↓}$ | atom $TV_↓$ | bond $TV_↓$ | bond len $W_{1↓}$ | bond ang $W_{1↓}$ |
|---|---|---|---|---|---|---|---|---|---|
| *data* | 99.9 | 99.9 | 99.8 | 100. | .001 | .001 | .025 | .000 | .050 |
| EDM | 40.3 | 97.8 | 87.8 | 99.9 | .285 | .212 | .048 | .002 | 6.42 |
| VoxMol$_{no\ rot}$ | 44.4 (±.1) | 96.6 (±.1) | 89.7 (±.2) | 99.9 (±.0) | .238 (±..001) | .025 (±.001) | .024 (±.001) | .004 (±.000) | 2.14 (±.02) |
| VoxMol | 75.0 (±1.) | 98.1 (±.3) | 93.4 (±.5) | 99.1 (±.2) | .254 (±.003) | .033 (±.041) | .036 (±.006) | .002 (±.001) | 0.64 (±.13) |
| VoxMol$_{oracle}$ | 81.9 | 99.0 | 94.7 | 97.4 | .253 | .002 | .024 | .001 | 0.31 |

Table 2: Results on GEOM-drugs. We use 10,000 samples from each method. Our results are shown with mean/standard deviation across 3 runs.

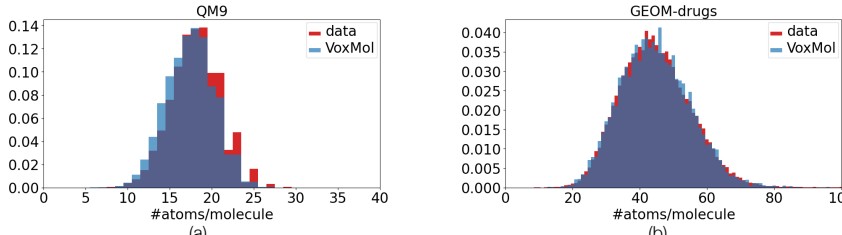

Figure 6: Empirical distribution of number of atoms per molecule on QM9 (left) and GEOM-drugs (right). We sample 10,000 molecules from train set and generate the same number of VoxMol samples.

to VoxMol$_{oracle}$ results, we see that VoxMol can still be vastly improved. We can potentially close this gap by improving the quality of the denoising network (*e.g.*, by improving the architecture, train on more data, efficient built-in SE(3)-equivariance CNNs, etc).

## 4.3  Ablation studies

**Noise level $\sigma$.**  Unlike diffusion models, the noise level is considered fixed during training and sampling. It is an important hyperparameter as it poses a trade-off between the quality of the walk step (Langevin MCMC) and the jump step (empirical Bayes). The ideal noise level is the highest possible value such that the network can still learn how to denoise. We train models on QM9 with $\sigma$ in $\{.6, .7, ..., 1.2\}$, while keeping all other hyperparameters the same. Figure 7(a,b,c) shows how noise level $\sigma$ influences the performance on the validation set. While most metrics get better as the noise level increases, others (like stable molecules and valency W1) get worse after a value. We observe empirically that $\sigma = .9$ is the sweet spot level that achieves better overall performance on the validation set of QM9.

**Number of steps $\Delta k$.**  Table 3 shows how VoxMol's performance on GEOM-drugs change with the number of walk steps $\Delta k$ on the Langevin MCMC sampling. In this experiment, we use the same trained model and only change the number of steps during sampling. Results of EDM are also shown for comparison (it always requires 1,000 diffusion steps for generation). We see that some metrics barely change, while others improve as $\Delta k$ increases. The average time (in seconds) to generate a

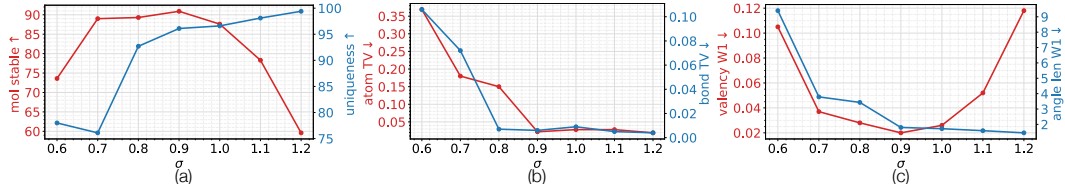

Figure 7: Effect of noise level $\sigma$ on generation quality. Models are trained on QM9 with a different noise level. Each plot shows two metrics: (a) molecule stability and uniqueness, (b) atom and bond TV, (c) valency and angle lengths W1.

| $\Delta k$ (n steps) | stable mol %$_\uparrow$ | stable atom %$_\uparrow$ | valid %$_\uparrow$ | unique %$_\uparrow$ | valency $W_{1\downarrow}$ | atom $TV_\downarrow$ | bond $TV_\downarrow$ | bond len $W_{1\downarrow}$ | bond ang $W_{1\downarrow}$ | avg. t s/mol.$_\downarrow$ |
|---|---|---|---|---|---|---|---|---|---|---|
| 50 | 78.9 | 98.7 | 96.3 | 87.8 | .250 | .073 | .102 | .002 | 1.18 | 0.90 |
| 100 | 78.6 | 98.6 | 95.5 | 94.3 | .256 | .050 | .101 | .002 | 1.62 | 1.64 |
| 200 | 77.9 | 98.4 | 94.4 | 98.6 | .253 | .037 | .104 | .002 | 1.02 | 3.17 |
| 500 | 76.7 | 98.2 | 93.8 | 99.2 | .252 | .043 | .042 | .002 | 0.56 | 7.55 |
| 1,000 | 75.5 | 98.4 | 93.4 | 99.8 | .257 | .029 | .050 | .002 | 0.79 | 14.9 |
| EDM | 40.3 | 97.8 | 87.8 | 99.9 | .285 | .212 | .048 | .002 | 6.42 | 9.35 |

Table 3: Effect of number of walk steps $\Delta k$ on generation quality on GEOM-drugs (2,000 samples). EDM results are shown for comparison.

molecule increases linearly with the number of steps, as expected. We observe that even using 500 steps, our model is still faster than EDM on average, while achieving better performance in these metrics. Remarkably, with only 50 steps, VoxMol already outperforms EDM in most metrics, while being *an order of magnitude* faster on average.

**Atomic density radii.** We also assess how the performance of the model changes with respect to the size of atomic radii chosen during the voxelization step (while always keeping the resolution of the grid fixed at .25Å). See appendix for how this is done. We tried four different values for the radii (same for all elements): .25, .5, .75 and 1.0. We observe—throughout different versions of the model, with different hyperparameters—that using a fixed radius of .5 consistently outperform other values. Training does not converge with radius .25 and quality of generated samples degrades as we increase the radius. We also tried to use Van der Waals radii (where each atom type would have their own radius), but results were also not improved.

## 5 Conclusion

We introduce VoxMol, a novel score-based method for 3D molecule generation. This method generates molecules in a fundamentally different way than the current state of the art (*i.e.*, diffusion models applied to atoms). The noise model used is also novel in the class of score-based generative models for molecules. We represent molecules on regular voxel grids and VoxMol is trained to predict "clean" molecules from its noised counterpart. The denoising model (which approximates the score function of the smoothed density) is used to sample voxelized molecules with walk-jump sampling strategy. Finally atomic coordinates are retrieved by extracting the peaks from the generated voxel grids. Our experiments show that VoxMol scales better with data and outperforms (by a large margin) a representative state of the art point cloud-based diffusion model on GEOM-drugs, while being faster to generate samples.

**Broader impact.** Generating molecules conditioned on some desiderata can have huge impacts in many different domains, such as, drug discovery, biology, materials, agriculture, climate, etc. This work deals with unconditional 3D molecule generation (in a pure algorithmic way): a problem that can be seen as an initial stepping stone (out of many) to this long-term objective. We, as a society, need to find solutions to use these technologies in ways that are safe, ethical, accountable and exclusively beneficial to society. These are important concerns and they need to be thought of at the same time we design machine learning algorithms.

**Acknowledgements.** The authors would like to thank the whole Prescient Design team for helpful discussions and Genentech's HPC team for providing a reliable environment to train/analyse models.

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

# A    Extra implementation details

## A.1    Voxel representation

Molecules in our datasets are converted into voxelized atomic densities. For each molecule, we consider a box around its center and divide it into discrete volume elements. We follow [93, 94] and first convert each atom (of each molecule) into 3D Gaussian-like densities:

$$V_a(d, r_a) = \exp\left(-\frac{d^2}{(.93 \cdot r_a)^2}\right),  \tag{5}$$

where $V_a$ is defined as the fraction of occupied volume by atom $a$ of radius $r_a$ at distance $d$ from its center. Although we could consider a different radius for each element, in this work we consider all atoms to have the same radius $r_a = .5$Å. The occupancy of each voxel in the grid is computed by integrating the occupancy generated by every atom in a molecule:

$$\text{Occ}_{i,j,k} = 1 - \prod_{n=1}^{N_a} \left(1 - V_{a_n}(||C_{i,j,k} - x_n||, r_{a_n})\right),  \tag{6}$$

where $N_a$ is the number of atoms in the molecule, $a_n$ is the $n^{th}$ atom, $C_{i,j,k}$ are the coordinates (i,j,k) in the grid and $x_n$ is the coordinates of the center of atom $n$ [93]. The occupancy takes the maximum value of 1 at the center of the atom and goes to 0 as it moves away from it. Every channel is considered independent of one another and they do not interact nor share volumetric contributions. We use the python package PyUUL [94] to generate the voxel grids from the raw molecules (`.xyz` or `.sdf` format).

We use grids with $32^3$ voxels on QM9 and $64^3$ on GEOM-drugs and place the molecules on the center of the grid. These volumes are able to cover over 99% of all points in the datasets. We use all 5 chemical elements present on the dataset (C, H, O, N and F), while for GEOM-drugs, we use 8 (C, H, O, N, F, S, Cl and Br). We model hydrogen explicitly in all our experiments. Finally, the input voxel grids are of dimension $\mathbb{R}^{5 \times 32 \times 32 \times 32}$ and $\mathbb{R}^{8 \times 64 \times 64 \times 64}$ for QM9 and GEOM-drugs, respectively. We augment the dataset during training by applying random rotation and translation to the molecules. For rotation, we sample three Euler angles uniformly (between $[0, 2\pi)$) and rotate each training sample. For translation, we randomly shift the center of the molecules on each of the three dimensions by sampling an uniform shift between $[0, .25]$Å.

## A.2    Architecture

Our neural network architecture follows standard encoder-decoder convnet architecture. We use a very similar architecture recipe to DDPM [34]. The model uses four levels of resolution: $32^3$ to $4^3$ for the QM9 dataset and $64^3$ to $8^3$ for the GEOM-drugs dataset. The input voxel is embedded into a 32 dimensional space with a grid projection layer (3D convnet with kernel size $3 \times 3 \times 3$). Each resolution (on both encoder and decoder) has two convolutional residual blocks. Each block contains a group normalization [95] layer, followed by SiLU [96] non-linearity and a 3D convnet (with kernel size $3 \times 3 \times 3$). All convolutions have stride 1 and we pad the feature maps with 1 on each side. We use self-attention layers between the convolutional layers in the two lowest resolutions. We reduce (increase, respectively) the resolution of the encoder (decoder) with $2 \times 2 \times 2$ (stride 1) max-poolings (bilinear-upsampling). The model has skip connections at each resolution to concatenate the encoder feature map with the decoder feature map. We double the number of feature maps at each resolution, except the last resolution where we quadruple. VoxMol has approximately 111M parameters. We also implemented a smaller version (with reduced number of channels per layer) with around 30M. These models achieve performance close to the base model and are faster to train and sample.

## A.3    Built-in SE(3) equivariance experiments

In early experiments, we made an attempt at using a SE(3)-equivariant 3D U-Net using steerable convnets [72] for denoising, but initial experiments were not successful. The hypothesis is that a built-in SE(3) equivariant version of our model, VoxMol$_{\text{equi}}$, would be advantageous over the non-equivariant version for the task of molecule generation. We start with the official implementation of [73] and tune several network hyperparameters (related to architecture, optimization and training) so that the network is able to achieve good denoising metrics on QM9. We then use the same procedure to

generate samples as described in the main paper (only switching the network from the non-equivariant to the equivariant version). We tried different sampling hyperparameters, but we were never able to achieve the same performance as non-equivariant VoxMol. Table 4 compares the results of the model with and without built-in SE(3) equivariance.

| QM9 | stable mol %$_\uparrow$ | stable atom %$_\uparrow$ | valid %$_\uparrow$ | unique %$_\uparrow$ | valency $W_{1\downarrow}$ | atom $TV_\downarrow$ | bond $TV_\downarrow$ | bond len $W_{1\downarrow}$ | bond ang $W_{1\downarrow}$ |
|---|---|---|---|---|---|---|---|---|---|
| VoxMol | 89.3 | 99.2 | 98.7 | 92.1 | .023 | .029 | .009 | .003 | 1.96 |
| VoxMol$_{equi}$ | 25.1 | 81.8 | 95.9 | 92.9 | 13.2 | .104 | .015 | .015 | 5.31 |

Table 4: Results on QM9 of our model without (VoxMol) and with (VoxMol$_{equi}$) built-in SE(3) equivariance.

There might be many reasons why this is the case: (i) the best reconstruction loss we found with the equivariant model is higher than the non-equivariant (approx. $9.4 \times 10^{-5}$ vs. $5.4 \times 10^{-5}$ MSE on val. set), (ii) the equivariant model needs more capacity to be competitive with the non-equivariant one (currently it has over $90\times$ fewer parameters), (iii) something in the sampling procedure needs to be different on the equivariant version (unlikely).

We hypothesize that if an equivariant version of VoxMol achieves similar (or lower) reconstruction loss as the vanilla version, it will probably achieve competitive/better results in the task of molecule generation. Finally, our equivariant implementation is less efficient (around 50-60% slower) and consumes more memory than the original version. This poses, therefore, an extra challenge to scale up the size of the dataset and the size of the molecules (*e.g.*, GEOM-drugs requires $64^3$ voxel grid).

## A.4 Training and sampling

The weights are optimized with batch size 128 and 64 (for QM9 and GEOM-drugs, respectively), AdamW optimizer ($\beta_1 = .9$, $\beta_2 = .999$), learning rate of $10^{-5}$ and weight decay of $10^{-2}$. The models are trained for 500 epochs on QM9 and around 24 epochs on GEOM-drugs. We discretize the underdamped Langevin MCMC (Equation 4) with the algorithm proposed by Sachs *et al.* [85] (this has been applied on images before [37]). Algorithm 1 describes this process.

---

**Algorithm 1:** Walk-jump sampling [1] using the discretization of Langevin diffusion by [85]. Lines 6-13 correspond to the *walk* step and line 14 to the *jump* step.

---

1: **Input** $\delta$ (step size), $u$ (inverse mass), $\gamma$ (friction), $K$ (steps taken)
2: **Input** Learned score function $g_\theta(y) \approx \nabla_y \log p(y)$ and noise level $\sigma$
3: **Output** $\widehat{x}_K$
4: $y_0 \sim \mathcal{N}(0, \sigma^2 I_d) + \mathcal{U}_d(0,1)$
5: $v_0 \leftarrow 0$
6: **for** $k = 0,...,K-1$ **do**
7: $\quad y_{k+1} \leftarrow y_k + \frac{\delta}{2} v_k$
8: $\quad g \leftarrow g_\theta(y_{k+1})$
9: $\quad v_{k+1} \leftarrow v_k + \frac{u\delta}{2} g$
10: $\quad \varepsilon \sim \mathcal{N}(0, I_d)$
11: $\quad v_{k+1} \leftarrow \exp(-\gamma\delta) v_{k+1} + \frac{u\delta}{2} g + \sqrt{u(1-\exp(-2\gamma\delta))}\varepsilon$
12: $\quad y_{k+1} \leftarrow y_{k+1} + \frac{\delta}{2} v_{k+1}$
13: **end for**
14: $\hat{x}_K \leftarrow y_K + \sigma^2 g_\theta(y_K)$

---

We use $\gamma = 1.0$, $u = 1.0$, $\delta = .5$ for all samplings and we generate multiple chains in parallel (200 chains for QM9 and 100 for GEOM-drugs). We follow [37] and initialize the chains by adding uniform noise to the initial Gaussian noise (with the same $\sigma$ used during training), *i.e.*, $y_0 = \mathcal{N}(0, \sigma^2 I_d) + \mathcal{U}_d(0,1)$ (this was observed to mix faster in practice).

All experiments and analysis on this paper were done on A100 GPUs and with PyTorch [97]. The models on QM9 were trained with 2 GPUs and the models on GEOM-drugs on 4 GPUs.

| dset | coords ref. | stable mol %↑ | stable atom %↑ | valid %↑ | unique %↑ | valency $W_1$↓ | atom TV↓ | bond TV↓ | bond len $W_1$↓ | bond ang $W_1$↓ |
|------|------|------|------|------|------|------|------|------|------|------|
| QM9 | - | 80.5 | 98.5 | 98.1 | 93.3 | .051 | .028 | .005 | .008 | 2.94 |
|     | ✓ | 89.3 | 99.2 | 98.7 | 92.1 | .023 | .029 | .009 | .003 | 1.96 |
| GEOM | - | 73.9 | 99.0 | 94.7 | 98.6 | .236 | .030 | .038 | .008 | 2.92 |
|      | ✓ | 74.9 | 98.1 | 93.4 | 99.2 | .254 | .033 | .036 | .002 | .63 |

Table 5: Effect of coordinate refinement on QM9 and GEOM-drugs. We use 10,000 samples from each method.

## A.5   Recovering atomic coordinates from voxel grid

Figure 4 shows our pipeline to recover atomic coordinates and molecular graphs from generated voxelized molecules. In the first step, we use the model to "jump" to the data manifold generating a sample in the voxelized representation, $x_k$. We set to 0 all voxels with value less than .1, *i.e.*, $x_k = \mathbb{1}_{\geq .1}(x_k)$. We then apply a simple peak finding algorithm to find the voxel coordinates corresponding to the peaks in the generated sample. Our peak finding algorithm uses a maximum filter with a $3 \times 3 \times 3$ stencil to find local maxima. Note that this algorithm always returns points on the voxel grid and is therefore limited by the resolution of the discretization.

In order to further refine the atomic coordinates we take advantage of the fact that our voxelization procedure is differentiable to perform gradient based optimization of the coordinates. Specifically we use L-BFGS to optimize the atomic coordinates based on the L2 norm of the reconstruction error in the voxel representation. Note, unlike some previous work [33] we perform peak detection and refinement in a single step and do not perform search over multiple possible numbers of atoms or atom identities.

Table 5 shows the effect of coordinate refinement on molecule generation. We generate molecules on the same setting in the experimental section.

Once we have obtained the optimized atomic coordinates, we follow previous work [33, 18, 17, 20] and use standard cheminformatics software to determine the molecule's atomic bonds.

## A.6   Metrics

Most of the metrics used to benchmark models come from [20][8]. Below we describe the metrics:

- *Atom stability:* the percentage of generated atoms with the right valency. This metric is computed on the raw 3D sample (before any postprocessing), therefore it is more stringent than validity.

- *Molecule stability*: the percentage of generated molecules where all its atoms are stable.

- *Validity:* The percentage of generated molecules that passes RDKit's sanitization filter.

- *Uniqueness:*. The proportion of valid molecules (defined above) that has a unique canonical SMILES (generated with RDKit) representation.

- *Atoms TV:* The total variation between the distribution of bond types in the generated and test set. We consider 5 atom types on QM9 and 8 atom types on GEOM-drugs. The histograms $\hat{h}_{\mathrm{atm}}$ and $h_{\mathrm{atm}}$ are generated by counting the number of each atom type on all molecules on both the generated and real sample set. The total variation is computed as:

$$\mathrm{Atoms\ TV}(\hat{h}_{\mathrm{atm}}, h_{\mathrm{atm}}) = \sum_{x \in \mathrm{atom\ types}} |\hat{h}_{\mathrm{atm}}(x) - h_{\mathrm{atm}}(x)|$$

- *Bonds TV:* Similar to above, the histograms for real and generated samples are created by counting all bond types on all molecules. The total variation is computed as:

$$\mathrm{Bonds\ TV}(\hat{h}_{\mathrm{bond}}, h_{\mathrm{bond}}) = \sum_{x \in \mathrm{bond\ types}} |\hat{h}_{\mathrm{bond}}(x) - h_{\mathrm{bond}}(x)|$$

---

[8]We do not compare directly with [20], since this model is an extension of EDM and leverages more information (connectivity graph and formal charges) during training.

- *Valency $W_1$:* This is the weighted sum of the Wasserstein distance between the distribution of valencies for each atom type:

$$\text{Valency W}_1(\text{generated,target}) = \sum_{x \in \text{atom types}} p(x) W_1(\hat{h}_{\text{val}}(x), h_{\text{val}}(x)),$$

where $\hat{h}_{\text{val}}(x)$ and $h_{\text{val}}(x)$ are the histogram of valencies for atom type $x$ for generated and holdout set samples, respectively.

- *Bond length $W_1$:* The weighted sum of Wasserstein distance between the distribution of bond lengths for each bond type:

$$\text{Bond Len W}_1(\text{generated,target}) = \sum_{b \in \text{bond types}} p(b) W_1(\hat{h}_{\text{dist}}(b), h_{\text{dist}}(b)),$$

where $\hat{h}_{\text{dist}}(b)$ and $h_{\text{dist}}(b)$ are the histogram of bond lengths for bond type $b$, for generated and holdout set samples, respectively.

- *Bond angles $W_1$:* The weighted sum of Wasserstein distance between the distribution of bond angles (in degrees) for each atom type in the dataset:

$$\text{Bond Ang W}_1(\text{generated,target}) = \sum_{x \in \text{atom types}} p(x) W_1(\hat{h}_{\text{ang}}(x), h_{\text{ang}}(x)),$$

where $\hat{h}_{\text{ang}}(x)$ and $h_{\text{ang}}(x)$ are the histogram of angles for atom type $x$ for generated and holdout set samples, respectively. See [20] for how angles are measured.

- *Strain energy:* The strain energy for a generated molecule is computed as the difference between the energy on the generated pose and the energy of a relaxed position. The relaxation and the energy are computed using UFF provided by RDKit. We use [91]'s implementation[9].

## A.7 Guiding the generation process

Like diffusion models, our method also leverages (learned) score functions and relies on Langevin MCMC for sampling. Therefore, in theory we can condition VoxMol similarly to how it is done in diffusion models: by constraining the score function as we walk through the MCMC chain. In the case of diffusion models, the score function of all steps is constrained to guide the transition steps from noise to a (conditioned) sample. In VoxMol, the constrained score function would affect the "walk steps" (the Langevin MCMC steps): it would restrict the region where the chain samples noisy molecules $y$ to $p(y|c)$ (instead of $p(y)$), $c$ is the condition (*e.g.*, gradient of a classifier). The "jump step" (a forward pass of the denoising network over the noised molecules) is independent of the condition and remains unchangeable.

Many of the innovations on conditioning diffusion models come from computer vision, where U-nets are usually used. Since VoxMol has the same architecture (albeit 3D instead of 2D), many of the conditioning techniques/tricks used in images may be more easily transferable. For example, we could in principle use the gradient of a classifier (trained jointly) to guide the sampling (using the same trick as in Dhariwal and Nichol [35]) or adapt gradient-free guidance ([34]). Pocket conditioning could also be possible, as in *e.g.*, [18, 67], but utilizing voxel representations instead of point clouds and neural empirical Bayes instead of diffusion models. In-painting has also proven to work very well in 2D U-Nets, so it could potentially work with 3D U-Nets as well. These in-painting techniques could also be leveraged in the context of molecule generation on voxel grids, *e.g.*, for linker generation, scaffold/fragment conditioning.

---

[9]Code taken from https://github.com/cch1999/posecheck/blob/main/posecheck/utils/strain.py

## B   Generated samples

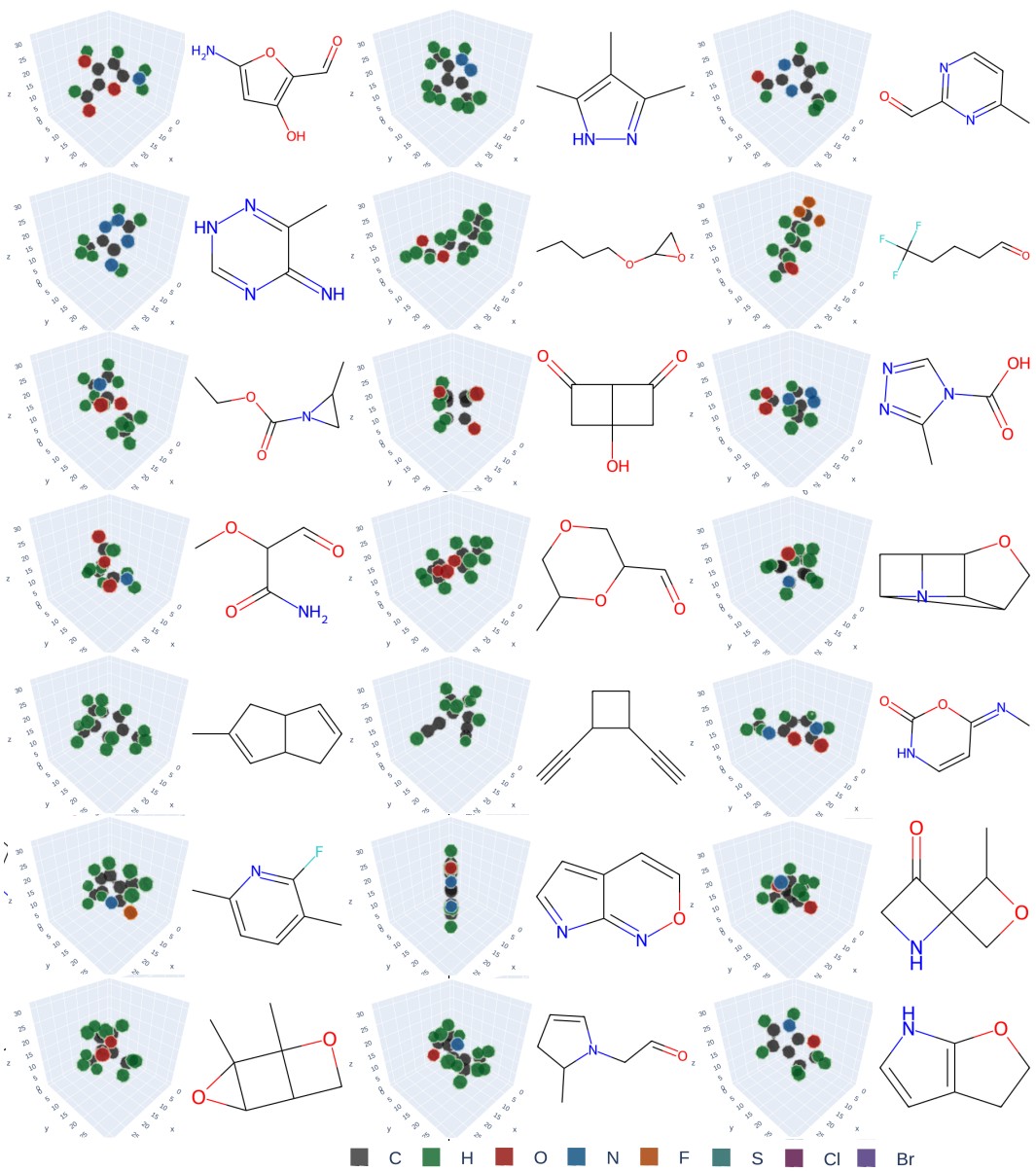

Figure 8: Random generated samples from VoxMol trained on QM9 (passing RDKit's sanitization). Molecular graphs are generated with RDKit.

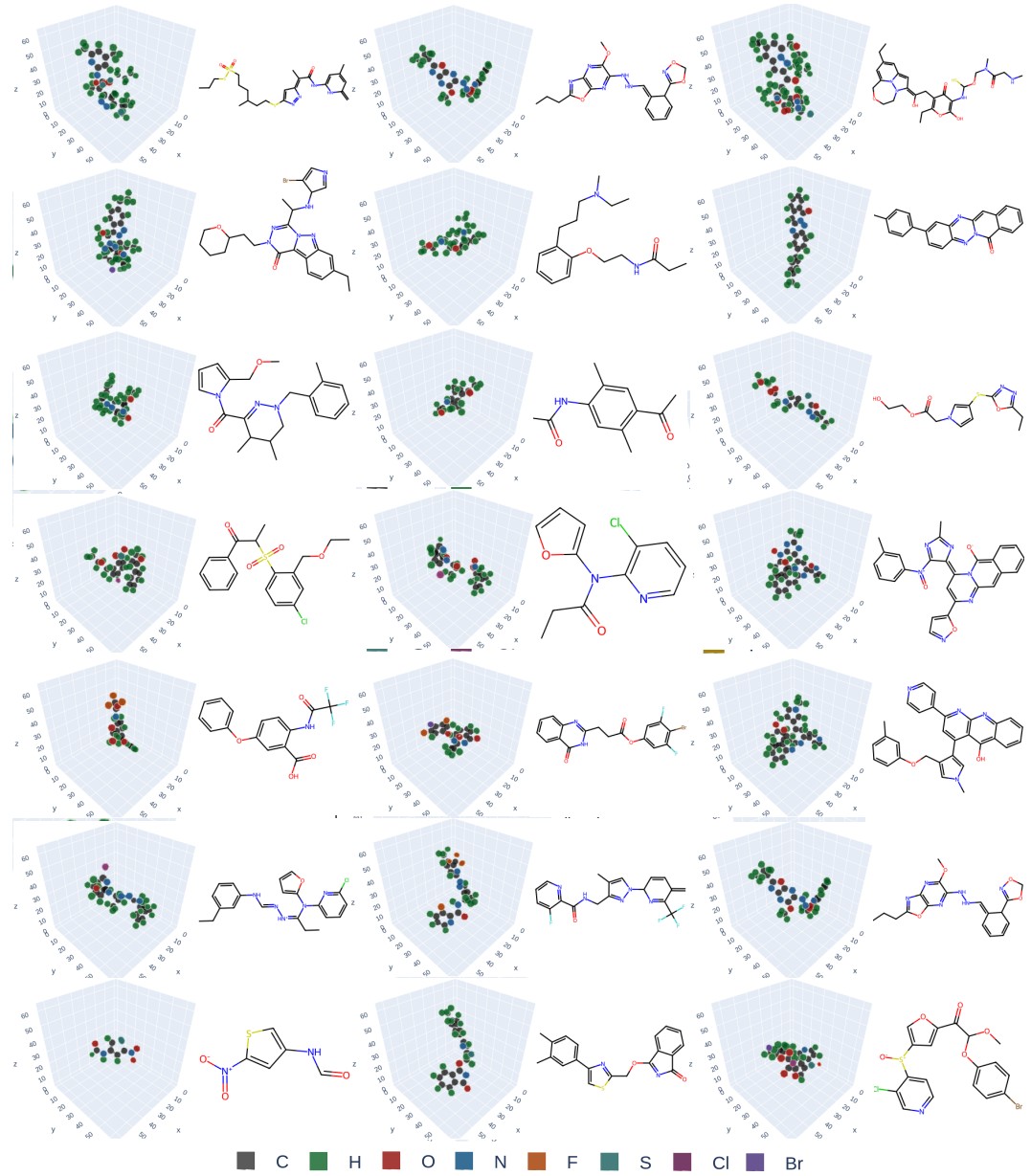

Figure 9: Random generated samples from VoxMol trained on GEOM-drugs (passing RDKit's sanitization). Molecular graphs are generated with RDKit.

