# OpenReview forum: "3D molecule generation by denoising voxel grids"
_NeurIPS.cc/2023/Conference — NeurIPS 2023 poster_

### Official Review · Reviewer_zHHZ · 2023-06-19

**Soundness:** 3 good
**Presentation:** 2 fair
**Contribution:** 2 fair
**Rating:** 6
**Confidence:** 4

**Summary:**

This paper describes a new method of unconditional small molecule generation by parameterizing small molecules as 3d voxel arrays.

**Strengths:**

- Novel characterization of the small molecule generation task and a new way to parameterize the data. This comes with benefits, chief among which is the ability to generate without a known number of atoms.

- I appreciate the discussion in section 4.3, where the authors dive into some problems with the metrics that they have chosen.

- The ablation studies are illuminating.

- The paper is well-written and easy to follow.

**Weaknesses:**

- Unconditional generation of small molecules is a hard task to evaluate. There are no good metrics to optimize for, as the paper indicates. Nevertheless, MiDi (work that the authors cite and compare against) has a more representative set of metrics they compare against. I suggest the authors do the same.

- Weaknesses of the one-step denoising procedure are not discussed. These include, for example, the inability to control generation through the use of auxiliary gradient signals, such as classifier-free guidance.

- The authors don't mention the implications of having a fixed grid size for their generative model. This means that the molecules generated need to be within a specified volume, restricting the output space. They should contrast this to the fixed atom number of competing models.

**Questions:**

- Line 243 should be as noise *decreases*, correct?

- We consistently see a large drop in the validity when the noise level goes from $\sigma = 0.9$ to $\sigma = 1$, why is such a small change in noise so detrimental? What are the characteristics of the generated molecules at these levels of noise? What happens if you train a model with $\sigma = 2$?

- What is the relevance of the number of aromatic rings per molecule? Especially given that this is one of the only metrics where the proposed model outperforms, I would like a good rationale for including this metric.

- What is the relevance of the validity metric, if that is something that can be automatically detected? It seems in this case that generating unique molecules is more relevant, especially if there are no other metrics to optimize.

- How can this model be extended to allow for conditional generation, given that most of the other methods being compared to have this ability?

**Limitations:**

Mainly, the authors should find better metrics to evaluate their work, such as those mentioned in the MiDi paper (at a minimum).

---

> ### Author Rebuttal · Authors · 2023-08-09
>
> We thank the reviewer for the feedback and suggestions. Below we address additional concerns from the reviewer.
>
> **Metrics:**
>
> We agree with the reviewers that unconditional molecule generation is hard to evaluate. We followed the reviewer's suggestion and evaluated VoxMol on the MiDi metrics (Table 1 and 2). VoxMol is slightly worse than EDM on the simple QM9 while performing much better than EDM on the more challenging/realistic GEOM-drugs dataset. We will update the tables and plots of the experiments to reflect the new metrics.
>
> **Limitations of one-step denoising:**
>
> Neural empirical Bayes (NEB) performs denoising in 1 step while diffusion models do in multiple (eg, 1000) steps.
> Because it is a single-step denoising process (from a high noise level), it can be challenging to capture high-frequency components of the signal (eg, generated images are usually more blurred than diffusion model samples). We observed empirically that VoxMol is able to denoise voxelized molecules (they are signals with much "lower frequencies"  than natural images), and therefore NEB is a good fit for this problem–as we show in experiments. We briefly mention this on L154-158, but will make it more explicit in the text.
>
> **Conditional generation:**
>
> Both NEB and diffusion models use a (learned) score function to sample with Langevin MCMC.. Therefore, we can in principle also guide the sampling by constraining the score function at each step of the MCMC chain.
> The Langevin MCMC is used very differently in the two approaches. In DM, the chains transition from pure noise to the molecule (that is, one chain per sample). In NEB, the chain keeps sampling noisy molecules from p(y) (and we get clean molecules by denoising them with xhat). Please see the paragraph on guided/conditional sampling on the main rebuttal.
>
> **No mention about the implications of fixed grid size:**
>
> With a fixed voxel grid, we are limited to a fixed volume in space (we mentioned this on L127, but will be more clear). With a 64^3 voxel grid (at resolution .25A), we can fit over 99% of the drug-like molecules from GEOM-drugs dataset. We wrote about the pros/cons of voxels vs point cloud (L100-117), but will be more specific on this tradeoff between max volume limitation/number of atoms.
>
> **Drop in validity from sigma=.9 to sigma=1:**
>
> As we increase sigma, denoising gets more and more difficult to learn. At the same time, the smoother the space of noisy molecules is (ie, higher the sigma), the easier it gets to sample from this space with Langevin MCMC. In practice, we need to find the highest sigma such that the U-Net can still learn to denoise. sigma=0.9 is the best value we found on our hyperparameter search (on the validation set). We also found this surprising; it seems that higher sigma values work on natural images (Saremi and Hyvarinen 19, Saremi and Srivastava, 22). We hypothesize this has to do with the sparsity of voxelized molecules compared to natural images, but it needs more investigation. We mention this in L174-176, but will provide a better discussion.
>
> **Measuring number of generated rings:**
>
> Hoogeboom et al., 22 (on appendix) points that predicting single bonds can easily increase validity. This metric was added to show that VoxMol achieves its performance without "cheating" and oversampling single bounds–in fact, it matches better the average number of rings from the ground truth dataset. Fig 8 shows that EDM undersample rings and oversamples single and double bonds, while VoxMol matches better the number of bonds. We recognize the rationale for this metric was not clearly explained in the text. We will move this discussion to the appendix of the paper and report the same metrics as in the MiDi paper, as suggested by the reviewers.
>
> **L243:** yes! Thanks for correcting.
>
> **Relevance of validity metrics:**
>
> We used validity metrics as they correlate with other metrics we measured. Following the reviewer's suggestion, we will report the MiDi metrics (Table 1 and 2 on the attached pdf) on the updated version of the paper.

---

> > ### Comment · Reviewer_zHHZ · 2023-08-14
> > **Increased score**
> >
> > I thank the authors for a detailed rebuttal. I believe the work is now stronger than when first presented. I have increased my score commensurate with the improved quality to a rating of accept.

---

> > > ### Author Response · Authors · 2023-08-18
> > >
> > > We thank the reviewer for the reviews and feedbacks. We particularly appreciate the suggestion of using the new set of metrics. We will incorporate the feedbacks on the updated version of the manuscript.

---

### Official Review · Reviewer_bpDM · 2023-07-03

**Soundness:** 2 fair
**Presentation:** 3 good
**Contribution:** 2 fair
**Rating:** 3
**Confidence:** 4

**Summary:**

This paper proposes VoxMol, a novel method for generating 3D molecules in the form of voxel grids. The proposed method adopts neural empirical Bayes as the basic probabilistic framework to develop generative models for 3D voxel grid representations of molecules. Experiments are conducted to show that the proposed method can successfully generate valid 3D molecules.

**Strengths:**

- This work proposes a novel method of generating 3D molecules in the form of 3D voxel grids by neural empirical Bayes framework. The exploration of generating 3D voxel grids representation of molecules is interesting and meaningful to the development of efficient drug discovery applications.
- The writing of this paper is well-organized and easy to follow.

**Weaknesses:**

- In this paper, generating 3D molecules in the form of 3D voxel grids by VoxMol is argued to be advantageous compred with 3D molecular graphs because the number of atoms is required to know beforehand, the distributions of continuous atom coordinates and discrete atom features are not treated separately, and simpler and faster training and generation can be achieved. However, in my opinion, these are not strong advantages.
(1) Actually, the number of atoms can be sampled from a learned distribution in 3D graph generation model, and VoxMol is also implicitly learn the number of atoms to place in every channel of voxel grids. These two processes are not fundamentally different.
(2) It is not a big issue to handle discrete and continuous distributions separately as we can jointly train discrete and continuous generative models (e.g., discrete & contiuous diffusion models). Also, VoxMol need to implicitly learn a "discrete distribution" as for every 3D spatial location in the voxel grid, it needs to decide the atom type, i.e., which channel to place an atom.
(3) The simpler and faster training and generation are mainly due to the use of neural empirical Bayes method, which may also be applied to 3D molecular graph generation to achieve fast generation and training. Also, 3D molecular graph generation can be accelerated by some other methods like introducing fragment-based generation.
Hence, it is not convincing that the proposed VoxMol method is really advantageous when compared to existing graph-based method.
- The use of 3D voxel grids also introduce many disadvantages. 3D voxel grids lose SE(3)-invariance as they are not invariant to rotations or translations. The use of 3D voxel grids hamper the introduction of chemical domain knowledge to guide the generative models to produce chemically valid or drug-like molecules, such as adding constraints based on fragments. The effect of these disadvantages should be evaluated or discussed.
- The experimental results in Table 1 do not demonstrate the advantages of the proposed VoxMol method as VoxMol does not outperform strong  baseline methods in several metrics.

**Questions:**

In experiments, methods are evaluated by the average number of aromatic rings per generated molecule. Do aromatic rings are detected from 2D molecule graphs inferred from 3D structures? Are the generated aromatic rings are checked in 3D space to decide whether they respect chemical constraints (i.e., all atoms in an aromatic ring should be in the same plane)? Why this metric is used? What can be shown if a method can generate more aromatic rings?

**Limitations:**

Yes.

---

> ### Author Rebuttal · Authors · 2023-08-09
>
> We thank the reviewer for the feedback and suggestions. Below we address additional concerns from the reviewer.
>
> **Implicitly learning the number of atoms is not an advantage:**
>
> We agree with the reviewer that VoxMol is implicitly learning the number of atoms (together with all other information about molecules). This might be arguable, but we assumed that learning representations more implicitly/end-to-end is simpler and can generalize better—therefore, we counted this as an advantage (R4 agrees: "This comes with benefits, chief among which is the ability to generate without a known number of atoms."). In applications related to in-painting (eg, pocket conditioning, linking, scaffold conditioning), learning implicitly the number of atoms could be very helpful. We will tone down this contribution by mentioning instead that VoxMol learns implicitly the number of atoms (in contrast to EDM-based methods, that need to learn them explicitly).
>
> **Score functions on continuous-only vs continuous/discrete space:**
>
> Because the score function is not defined on discrete spaces, it can be non-trivial to adapt the score-based modeling to categorical data. This is currently an active research area (eg [E-H]) and we still don't have a clear winner solution to the problem. It has also been pointed ([F]) that it can be more challenging to condition the generation of score-based models on discrete spaces. Therefore, as we agree with the reviewer that it is possible to learn score function on continuous-discrete space, we believe that training only on continuous space is simpler.
>
> **Simpler training because to NEB is simpler than diffusion:**
>
> We agree with the reviewer. In fact, we mention in the text that VoxMol (based on NEB) is simpler to train than _point-cloud diffusion models_ (L54-55 and L154), but not a general advantage of voxels over point clouds.
> We also point out that NEB requires a very large noise to work and denoising point clouds can be more challenging than denoising discrete grids (U-Nets works extremely well for this task and has been highly fine-tuned for this task with large image generation models).
>
> **Experimental results:**
>
> The main conclusion from our experiments is that VoxMol underperforms EDM by a small margin on QM9 (Table 1) and outperforms EDM on GEOM-drugs (larger/more realistic dataset) by a larger margin (Table 2). This conclusion holds even further with the MiDi metrics suggested by the reviewers (see Table 1 and 2 on the attached pdf).
>
> **"3D voxel grids lose SE(3)-invariance as they are not invariant to rotations or translations."... "should be discussed"** [we assume the reviewer means "equivariance" instead of "invariance". Please let us know if that is not the case]:
>
> Our method has built-in equivariance to translation and relies on data augmentation through rotation to learn rotation equivariance (as it has been done before). We mentioned in the manuscript that VoxMol (being based on CNNs) does not have rotation equivariance built-in in the architecture (L113-117 and L217-218). For clarity, we will include an explicit section mentioning the limitations of the model on the revised version of the paper.
>
> **Guide generation:**
>
> Adding constraints to guide VoxMol generation can in principle be done in the same way as in-painting or guidance is done with diffusion models on images (eg, Song et al. 20, Lugmayr et al. 22). Please see general rebuttal for discussion in guided/conditional sampling.
>
> **Measuring number of generated rings:**
>
> Hoogeboom et al., 22 (appendix) points that predicting single bonds can easily increase validity. This metric was added to show that VoxMol achieves its performance without "cheating" and oversampling single bounds–in fact, it matches better the average number of rings per molecule from the ground-truth dataset. Fig 8 in the submission shows that EDM undersample aromatic bonds and oversamples single and double bonds, while VoxMol matches better the number of bonds. We recognize the rationale for this metric was not clearly explained in the text. We will move this discussion to the appendix of the paper and report the same metrics as in the MiDi paper, as suggested by the reviewers.
>
> [E] Argmax Flows and Multinomial Diffusion, Hoogeboom et al, Nuerips21
>
> [F] Continuous diffusion for categorical data, Dieleman et a., 22
>
> [G] Analog bits, Chen et al., ICLR23
>
> [H] Geometric Latent Diffusion Models for 3D Molecule Generation, Xu et al, ICML23

---

> > ### Comment · Reviewer_bpDM · 2023-08-12
> > **Follow-up Responses**
> >
> > I appreciate authors' hard work in the rebuttal. While some of my concerns have been addressed, several key concerns still remain unsolved to me.
> >
> > - I agree that EDM cannot implicitly learn the number of atoms and implicitly learning it is useful in some conditional generation applications. But graph-based autoregressive generation methods can also achieve it by generating a "STOP" token to indicate the stop of sequential generation while not specifying the number of atoms at start. As authors also agree that jointly learning continuous & discrete distributions are not impossible and simpler training comes from NEB instead of image-based training, I suggest to remove the strong claim that image-based generation is better than graph-based generation in the paper revision, but simply treat the image+NEB as an independent method and discuss its unique advantages.
> > - I agree that the number of aromatic rings is a meaningful metric. However, as authors mentioned, aromatic rings are identified by aromatic bonds, so how the aromatic bonds are detected from the generated 3D molecular structures? Are they detected by by interatomic distances? However, even generating a 6-aromatic-bond cycle graph is not enough for an aromatic ring, as by chemistry knowledge, in a valid aromatic ring, all atoms (6 carbon and 6 hydrogen atoms) should be in the same plane. Do authors check if this constraint is satisfied?
> > - Authors are recommended to discuss the effect of hampering fragment-based generation. Here, fragments are those functional groups or subgraphs frequently appeared in molecules, such as aromatic rings or alkene. By chemical knowledge, the 3D structures of these functional groups have many constraints, e.g., containing some unrotatable bonds that fix some atoms in the same plane. These chemical constraints can be well incorporated into graph-based generation methods, as we can first generate 3D "super-node" graph where each super node is a functional group, then refine the 3D atomic structures in each super node (an example of this method is [1]). However, image-based methods cannot achieve it in my opinion, as images separate atoms in a functional group into different channels in a 3D image volume. This leads to a major concern that molecules generated in the form of 3D images may consistently break chemical constraints and are not practically synthesizable.
> >
> > Due to the above concerns, I tend to keep my decision of rejection.
> >
> > [1] Coarse-to-Fine: a Hierarchical Diffusion Model for Molecule Generation in 3D. ICML 2023.

---

> > > ### Author Response · Authors · 2023-08-13
> > >
> > > We thank R3 for the prompt reply.
> > > We provide additional discussions in the hope of better answering R3's questions.
> > >
> > > Below, we clarify the three concerns raised by the reviewer (see below for detailed clarifications):
> > > - **Point 1:** we agree with each other. As suggested, we will remove the three initial claims and focus on the model's advantages only.
> > > - **Point 2:** we agree the # arom. rings metric is not very informative (we use RDKit to compute it).  We follow reviewers suggestions and use MiDi metrics, as it is better for benchmarking. Table 2 on the attached pdf shows that our model clearly beats EDM by a large margin.
> > > - **Point 3:** voxel-based models can in principle do fragment-based generation/have chemical constraints (eg, [A, B, C] do this). However, the contributions of this work are (i) proposing a new model for unconditional generation and (ii) beating the SOTA on a large/challenging dataset on unconditional generation. Fragment-based generation is beyond the scope of this paper and can be seen as future applications (same for structure-based or scaffold-based generation).
> > >
> > > We hope the clarifications above—together with results on GEOM-drugs (better than EDM by a large margin)—can change the reviewer's opinion. If not, please let us know where we can be more clear.
> > >
> > > Are these three points all the concerns the reviewer have? We would be happy to address any other concern the reviewer might have.
> > >
> > > ---
> > > Detailed comments:
> > > - **Comparison with autoregressive (AR) models:**  when we mention about learning the #atoms implicitly, we were comparing VoxMol with diffusion models (current SOTA) and not with AR models. AR models have their own advantages/disadvantages (eg, it has not been successfully applied on GEOM-drugs).
> > > - **Continuous/discrete spaces:** authors and reviewer agree on this.
> > > - **Number aromatic ring:**  we computed the number of rings per molecule with RDKit (it is a simple “2D metric”, computed on valid molecules, like many of the other metrics used in this task) and therefore it does not check if atoms are on the sample plane (we observed qualitatively that they do many times). We agree this is not the most meaningful metric and will not report it. Table 1 and 2 on joint pdf show results on MiDi metrics that are more useful for benchmarking models (proposed by reviewers).
> > > - **Fragment-based generation**:
> > >   - Voxel-based models can in principle do fragment-based generation/incorporate chemical knowledge. These constraints can be imposed either (i) during sampling (similar to in-painting, eg, initializing the chain with fragments and keeping them unchanged during sampling) or (ii) apply fragment constraints during training.
> > >   - _“However, image-based methods cannot achieve it to me, as images separate atoms in a functional group into different channels in a 3D image volume.”_:  Previous works have shown evidence that this is possible (eg [A, B, C]). In fact, some chemical priors like 3D pharmacophores are particularly easy to incorporate with voxels.
> > >   - This paper is about proposing a new model for unconditional generation and beating the SOTA on a large/challenging dataset. Fragment-based generation is beyond the scope of this paper.
> > >
> > > [A] DeepFrag: a deep convolutional neural network for fragment-based lead optimization, Green et al, 21
> > >
> > > [B] Deep generative design with 3D pharmacophoric constraints, Imrie et al, , 21
> > >
> > > [C] Incorporating target-specific pharmacophoric information into deep generative models for fragment elaboration, Hadfield et al, 22

---

> > > > ### Comment · Reviewer_bpDM · 2023-08-17
> > > > **Response**
> > > >
> > > > I appreciate authors' response. However, my concerns have not been fully addressed.
> > > >
> > > > - As authors agree that the original claimed advantages of image-based generation are not persuasive, it seems that there are not outstanding advantages of the proposed method over existing graph-based 3D molecule generation methods. Hence, I think authors should discuss what are the outstanding and unique advantages of the proposed method in the revision of paper, otherwise I am worried that the contribution of this work does not reach the bar of top-tier conferences like NeurIPS.
> > > > - I do not think fragment-based generation is beyond the scope of this work. This work studies unconditional generation of 3D molecules, and this problem is tightly related to real-world molecule design application. In real-world applications, we have to consider whether the designed 3D molecules satisfy chemical constraints, because a 3D molecule violating these constraints will even not stably exist in nature, let alone be synthesized. This is the central reason why I emphasize fragments, because many chemical constraints are in the level of fragments or functional groups. For graph-based generation, chemical constraints can be intuitively satisfied by considering fragments as super nodes and using coarse-to-fine generation, but in contrast, image-based generation seems not to have an intuitive way to satisfy chemical constraints. But authors have pointed out some papers about this topic, hence, in the revision of paper, I suggest authors to discuss the possibility of introducing chemical constraints or other domain knowledge into the proposed method so as to improve the synthesizability of the generated molecules.
> > > > - It is nice to see that VoxMol can outperform EDM on GEOM-Drugs by a large margin. I suggest to add an invariant of EDM [1] as an additional baseline in the revision of paper.
> > > >
> > > > [1] Diffusion-based Molecule Generation with Informative Prior Bridges. NeurIPS 2022.

---

> > > > > ### Author Response · Authors · 2023-08-18
> > > > >
> > > > > We really appreciate the engagement of the reviewer. Below we provide additional discussions in the hope of better answering R3's questions.
> > > > > - **"There are not outstanding advantages of the proposed method over existing graph-based 3D molecule generation methods":**
> > > > >   - There are already many papers showing the usefulness of voxel-based representations for molecules (in discriminative, unconditional generation, conditional generation on pockets, scaffolds, fragments, etc). This is not the contribution of this work.
> > > > >   - The advantages of our model are the following:
> > > > >      - The proposed model is fundamentally different from the current SOTA. It has different data representation, generative modeling algorithm, noise model and the neural network used.
> > > > >     - We show for the first time a voxel-based model can be competitive with graph-based on molecule generation on standard benchmarks.
> > > > >     - We show it can scale better to larger datasets,  achieving better performance on the challenging GEOM-drugs.
> > > > >   - The proposed method is so different from SOTA that it "shed light on new research directions of connecting 3D CNNs to equivariant GNNs" (as pointed out by R2).
> > > > >
> > > > > - **"Discuss what are the outstanding and unique advantages of the proposed method":**  We clearly state the unique advantages of our method on the abstract (L7-12, and throughout the text): `Our method, VoxMol, generates molecules in a fundamentally different way than current state of the art (i.e., diffusion models applied to atom point clouds). They differ in terms of the data representation, the noise model, the network architecture and the generative modeling algorithm. VoxMol achieves comparable results to state of the art on unconditional 3D molecule generation while being simpler to train and faster to generate molecules.`
> > > > >
> > > > > - **"I do not think fragment-based generation is beyond the scope of this work.":**
> > > > >   1. We agree that conditional molecule generation is the more important task—not only on fragments, but also on pockets, scaffolds, etc. Since this model is the first of its kind, we first study the unsupervised generation case to understand its capacities/limitations.
> > > > >   2. No other work on unconditional generation that we know of--including [1] mentioned by R3– provides information on doing fragment-based generation (as this is beyond what papers want to show).  In fact, the only work that mentioning fragments is Vignac et al22: _"In contrast to the proposed method, which operates at the node level, fragment-based methods [13, 24, 32] learn to combine chemically-relevant substructures from a fixed or learned dictionary [48], but are harder to adapt to 3D."_
> > > > >   3. We will add a paragraph mentioning voxel-based models can in principle do fragment-based generation. Given that (i) there are published works that show that voxel-based models can condition on fragments (and pockets, and scaffolds) and (ii) point 2 above, we argue that providing results on fragment-based generation should not be a requirement for acceptance.
> > > > >
> > > > > - **Comparison with [1]:**
> > > > >   - We cited this work in the manuscript but direct comparison is hard. The authors do not provide code (nor generated samples).
> > > > >   - Table 1 of [1] shows that their model achieves 82.4/0.0%, on atom and molecule stability, respectively, on GEOM-drugs (they only report those two numbers). In their setup, their performance is 1% better than EDM. However, we can't rely on comparisons, as [1] does not provide details on how training data is preprocessed or how evaluation is done in detail.
> > > > >
> > > > > Were we clear on the unique advantages of our model and why we think fragment-based conditioning should not be a requirement for acceptance?
> > > > > Please, let us know if any other clarification is necessary.

---

> > > > > > ### Comment · Reviewer_bpDM · 2023-08-18
> > > > > > **Response**
> > > > > >
> > > > > > I appreciate authors' response to address my concerns in follow-up responses. Here are my responses.
> > > > > >
> > > > > > - For the advantages, I am not satisfied with authors' responses. What I would like to know is the unique advantages that exist in VoxMol but does not exist in current graph-based generation methods. But authors' responses emphasize the differences between VoxMol and graph-based methods in methodology design. Differences are not advantages. "First time to show a voxel-based model can be competitive with graph-based on molecule generation on standard benchmarks" is not enough, because I want more insightful analysis about why VoxMol is competitive with (or has the potential to significantly outperform) graph-based methods from the perspective of data representation and model design.
> > > > > > - Specifically, I would like to know answers to these questions --- now I want to develop a drug discovery system, why I choose your image-based molecule generation method given that I already have many existing graph-based molecule generation methods as candidates (not only EDM but also some other better options like [1][2][3])? Compared with graph-based molecule representation, can image-based molecule representation in VoxMol improve the quality of generated molecules so that their 3D structures respect chemical constraints better, especially fragment-level constraints?
> > > > > > - In authors' responses, it is highlighted that VoxMol achieves better performance than strong graph-based baseline EDM on large benchmark GEOM-Drug. Maybe Authors will claim it as answers to the above questions. However, the "better performance" comes from higher "Validity" metric value, which may not be a reliable metric to me. To my understanding, "Validity" metric bascially converts 3D structure to 2D molecular graphs and check if bond valencies are correct. If this is true, it does not check the quality of 3D structures in a "fragment" view. For instance, the converted 2D molecular graphs may show that you have generated a beneze ring but 3D coordinates of corresponding atoms are not in a plane, then this is not a chemically valid beneze ring but "Validity" metric still counts it as "valid". In this case, good "Validity" metric does not necessarily guarantee that the generated molecules can stably exist in nature and synthesized.
> > > > > > - Note that I am not talking about fragment-based generation as something like generating conditioned on some fragments. I mean composing molecules by a set of fragments, such as taking a beneze ring and a carboxyl to form a new molecule. Actually, fragment-based generation have been tried by researchers in molecule generation, e.g., [1][4] ([1] is unconditional generation and [4] is conditional generation from protein pockets, but it is not very different from unconditional generation). Particularly, [1] achieves very good "Validity" metric shown in Table 6, and more importantly, [1] incorporates many prior chemical knowledge into 3D fragment structures so it significantly increases the chances that the generated molecules can stably exist in nature and synthesized. As [1] is published very recently, I do not require to compare with [1] but I expect authors to know that fragment-based generation has already been explored.
> > > > > > - To summary, I think the most important thing is to give good answers to the questions in my second point --- find advantages that uniquely exist in image-based molecule representations and discuss incorporating chemical constraints into image-based molecule representations. Given there already exist many literatures and new methods in unconditional molecule generation now, I do not think simplying proposing a novel method and achieving good metric values in benchmarks (particularly "Validity" metric is not so reliable to me) are solid technical contributions. I will not think this paper reaches the acceptance bar unless questions are well answered.
> > > > > >
> > > > > > [1] Coarse-to-Fine: a Hierarchical Diffusion Model for Molecule Generation in 3D. ICML 2023.
> > > > > > [2] Diffusion-based Molecule Generation with Informative Prior Bridges. NeurIPS 2022.
> > > > > > [3] Geometric Latent Diffusion Models for 3D Molecule Generation. ICML 2023.
> > > > > > [4] Molecule Generation For Target Protein Binding with Structural Motifs. ICLR 2023.

---

> > > > > > > ### Author Response · Authors · 2023-08-18
> > > > > > > **On advantages of vision-based approaches over graph-based ones for molecule generation**
> > > > > > >
> > > > > > > We appreciate the continued discussion with the reviewer as we believe it further clarifies the strengths of our paper. We would like to summarize our responses earlier here:
> > > > > > >
> > > > > > > - The main advantage of a vision-based approach to molecule generation is **expressivity**. Graph-based methods suffer there due to the message passing formalism, a very well-known problem in the literature [XHLJ19]. In particular, we use the U-Net architecture with attention heads that are currently the workhorse behind state-of-the-art (score-based) generative models on images. With more data and more computation, these models can only become better.
> > > > > > >
> > > > > > > - Related to the above comment and highlighting our results, we have already seen much better performance on the larger GEOM-drugs dataset compared to smaller QM9. We suspect this gain will become even more pronounced for (future) larger datasets; this is not a blind suspicion, but rooted in limitations of graph-based message-passing frameworks when it comes to expressivity. "Respecting constraints" has less of a value if one can not capture the modes of a distribution with high fidelity. We agree with the reviewer that current metrics have faults but respecting chemical constraints is not the final answer either.
> > > > > > >
> > > > > > > - Another advantage is that what we propose complements the point-cloud/GNN approaches. On this point we agree with **Reviewer j1qM**  that
> > > > > > > >"the paper showcases that instead of the traditional graph representation, molecule generation can also be performed on 3D voxels, shedding light on new research directions of connecting 3D CNNs to equivariant GNNs".
> > > > > > >
> > > > > > > - The reviewer's practical question on developing a "drug discovery system" is particularly interesting from this angle. We have shown a clear benefit in having an expressive model that's also efficient with only a *single* hyperparameter, the noise level $\sigma$. The question if we can "respect chemical constraints better" is really a research question. We think about it from a different angle: in our view, respecting chemical constraints is akin to respecting grammar in language modeling. Current state-of-the-art models do not enforce grammar but with enough data and expressive architectures (like transformers) this has become a moot issue. We have made a step in the right direction in demonstrating the power of a score-based computer vision approach for this important practical problem.
> > > > > > >
> > > > > > > - Specifically, on the question of  *“why I choose your image-based molecule generation method given that I already have many existing graph-based molecule generation methods as candidates (not only EDM but also some other better options like [1][2][3])?”* We believe if the choice is binary the answer will depend on the dataset. We have made a strong case in this paper that our vision-based approach is a viable candidate for large datasets. We have also seen its limitations on small datasets, where having implicit priors are indeed beneficial. This again goes to the heart of this discussion, there are benefits to both having strong priors imposed on the architecture and there are also benefits for having expressive architectures. We believe for larger datasets the latter becomes even more important.
> > > > > > >
> > > > > > > - We agree with the reviewer regarding the validity metric, but we have outperformed a strong baseline such as EDM on _eight out of nine_ metrics on GEOM-drugs (see Table 2 on rebuttal pdf)--the dataset that matters for drug discovery going back to the reviewer's earlier question. E.g., 70% of Voxmol generated molecules are stable (meaning all atoms in the mol have right valence), while this degrades to 40% on EDM; VoxMol also captures much better the distribution of atoms (EDM highly underrepresents Carbon), valencies and bond types and angles.
> > > > > > >
> > > > > > > Again, we very much appreciate this discussion, and we hope the reviewer has a clearer picture of the advantage of our vision-based approach to molecule generation vs. graph-based ones. We hope the reviewer takes our clarifications and our strong results on GEOM-drugs into account and will support our paper for acceptance.
> > > > > > >
> > > > > > > [XHLJ19] K Xu, W Hu, J Leskovec, S Jegelka, *How powerful are graph neural networks?* ICLR 2019.

---

> > > > > > > > ### Comment · Reviewer_bpDM · 2023-08-19
> > > > > > > > **Response**
> > > > > > > >
> > > > > > > > I appreciate authors' response, but my questions still remains unsolved.
> > > > > > > >
> > > > > > > > - I agree that image-based generative models can take the advantage of expressive models like U-Net. However, it is not appropriate to underestimate the expressivity of graph models. Actually, graph models for 3D molecules or point clouds can be made much more expressive by using high-rotation-order features [1], atomic cluster expansion mechanism [2], or equivariant transformer architecture [3]. Graph-based generative models can become as expressive as image-based generative models by using these networks as backbones.
> > > > > > > > - Even if graph models do not reach the same expressivity as vision models, they are more data efficient as they are internally designed to satisfy SE(3)-equivariance. I agree that if sufficient amounts of data are provided, vision models can also capture SE(3)-equivariance, chemical constraints, or other "grammars", just like large language models. However, while it is relatively easy to collect large amounts of language data, it is very expensive to collect the same amount of 3D molecule structure data, as producing 3D molecule structures need expensive computation like DFT. Considering this practical difficulty, graph models become more advantageous due to its data efficiency.
> > > > > > > > - I do not agree that "Respecting constraints has less of a value if one can not capture the modes of a distribution with high fidelity". Developing unconditional generation methods of 3D molecules must consider how the methods will be applied to real-world applications, and respecting chemical constraints is a requirement as a 3D molecule that cannot stably exist in nature or be synthesized is of little use. Though respecting chemical constraints may limit the flexibility of generation models, I believe they cannot be neglected.
> > > > > > > >
> > > > > > > > [1] Geometric and Physical Quantities Improve E(3) Equivariant Message Passing. ICLR 2022.
> > > > > > > > [2] MACE: Higher Order Equivariant Message Passing Neural Networks for Fast and Accurate Force Fields. NeurIPS 2022.
> > > > > > > > [3] Equiformer: Equivariant Graph Attention Transformer for 3D Atomistic Graphs. ICLR 2023.

---

> > > > > > > > > ### Author Response · Authors · 2023-08-21
> > > > > > > > >
> > > > > > > > > We thank the reviewer for their follow-up, and we appreciate that we agree on the unique advantages of our model concerning the issue of expressivity. Many years of research have gone into the design of neural architectures in computer vision and our approach readily takes advantage of these developments.
> > > > > > > > >
> > > > > > > > > We believe the remaining disagreements do not seem to be on the merits of the paper. Our work is the very first score-based generative model for molecule generation that is vision based. The paper is novel with regards to the methodology as acknowledged by the reviewer. In our view, the work is complete in its scope: it is grounded in theory, and we have also shown very strong performance on a large drug-like dataset (GEOM-drugs).
> > > > > > > > >
> > > > > > > > > We hope the reviewer focuses more on our agreements than disagreements on the issue of SE(3) equivariance. We also believe this somewhat philosophical discussion (we do not mean to diminish its importance) should not be the reason to reject the paper.

---

> > > > > > > > > > ### Comment · Reviewer_bpDM · 2023-08-21
> > > > > > > > > > **Response**
> > > > > > > > > >
> > > > > > > > > > Actually, I do not agree that expressivity can be considered as a unique advantage because graph-based generation methods can also take the advantage of many expressive models as I mentioned in my last response. Also, I do not agree that issues of data efficiency and respecting chemical constraints are not the metrits of the paper, because they are unneglectable problems in real-world applications. As my major concerns remains unsolved yet, I will keep my rating.

---

> > > > > > > > > > > ### Author Response · Authors · 2023-08-21
> > > > > > > > > > >
> > > > > > > > > > > The score of 3 is reserved for *"a paper with technical flaws, weak evaluation, inadequate reproducibility and incompletely addressed ethical considerations."* In this process, the reviewer has not brought a single instance of "technical flaw", "weak evaluation", or " inadequate reproducibility".
> > > > > > > > > > >
> > > > > > > > > > > Please note that expressivity is not something one can *measure*. The reviewer acknowledged earlier that  "I agree that image-based generative models can take the advantage of expressive models like U-Net" but the discussion has turned into proving the "unique" advantage of our model. As we mentioned earlier, this discussion (and the reviewer's score as it appears to be the case) is not on the merits of the paper.
> > > > > > > > > > >
> > > > > > > > > > > Regarding the expressive power of message-passing formalism and graph neural networks, we have already cited the influential paper *"How powerful are graph neural networks?"* in our earlier response. Higher order message passing can alleviate this problem to a degree (dictated by how "higher order" one can computationally afford), but they come at a cost and they have been limited to third-order message passing frameworks in practice including in the paper *"MACE: Higher Order Equivariant Message Passing Neural Networks for Fast and Accurate Force Fields"* the reviewer referred to earlier.

---

### Official Review · Reviewer_j1qM · 2023-07-04

**Soundness:** 3 good
**Presentation:** 3 good
**Contribution:** 3 good
**Rating:** 6
**Confidence:** 4

**Summary:**

This paper proposes a novel 3D molecule generation routine dubbed VoxMol. The highlight of it lies in its introduction of a connection between traditional molecular graph representation and 3D voxel representation. In VoxMol, the molecules are first translated into voxel representation. A denoising network is then trained to serve as a score approximation. In the sampling process, a novel walk-jump procedure is introduced to sample noisy p(y) and obtain the denoised x.

**Strengths:**

1. To my knowledge, the method is novel in terms of data representation and the training/sampling process. The paper showcases that instead of the traditional graph representation, molecule generation can also be performed on 3D voxels, shedding light on new research directions of connecting 3D CNNs to equivariant GNNs. The authors also tried to recover the molecular graph representation based on the denoised voxel, which I really appreciate.

2. The evaluations are interesting. Though not always state-of-the-art, the authors have exhibited insightful investigations, e.g., how $\sigma$ influences the tradeoff between validity and uniqueness.

3. The overall presentation is clear and the method is very easy to follow.



**Weaknesses:**

1. I feel a little pity that equivariance is not properly injected into the model. I think it would make a much stronger impact if equivariance could be enforced even on this 3D voxel representation so that physical/chemical priors could be ensured.

2. The method does not significantly beat EDM on these benchmarks. But I understand that as perhaps the first work to challenge the task of 3D molecule generation with 3D CNNs, it is difficult to beat all other GNN counterparts at once.

**Questions:**

As the authors have stated in the paper, efforts have been made for equivariant 3D CNNs but were unsuccessful. I am curious about what detailed experiments have been conducted and how were the results. I believe by involving equivariance the work will make a stronger contribution.

**Limitations:**

The limitations are not discussed in the main text. I would recommend the authors add these discussions since I think the work is inspiring and will benefit more from future research.

---

> ### Author Rebuttal · Authors · 2023-08-09
>
> We thank the reviewer for the feedback and suggestions. Below we address additional concerns from the reviewer.
>
> **The method does not significantly beat EDM on these benchmarks:**
>
> Table 1 and 2 on the attached pdf compares our method with EDM using MiDi metrics (as suggested by reviewers). Although it underperforms EDM on QM9 (by a relatively small margin), Voxmol clearly beats EDM by a large margin on most metrics on the challenging (and more realistic!) GEOM-drugs dataset. These conclusions were the same with the original metrics, but the new metrics show the differences in performance more clearly. We will update the tables and plots of the experiments to reflect the new metrics.
>
> **Limitations:**
>
> We compare the pros/cons of voxel/CNN vs point-cloud/GNN on the last paragraph of related works (L100-117). We also mention the limitation that it scales cubic with voxel grid dimension (although been constant time given a fixed grid dimension) on L257-259. We would make these limitations more clear by having an explicit "Limitations" paragraph at the end of the manuscript (and also updating with reviewers' remarks).
>
> **SE(3) equivariance:**
>
> Please see SE(3) equivariance paragraph in general rebuttal.
>
> **SE(3)-Equivariant 3D CNN experiments:**
>
> We start with the official implementation of [D] and tune several network hyperparameters (related to architecture, optimization and training) so that the network is able to achieve good denoising metrics on QM9.  We then use the same procedure to generate samples as described in the paper (only switching the network from the non-equivariant to the equivariant version). We tried different sampling hyperparameters, but we were never able to achieve the same performance as non-equivariant VoxMol (eg, the molecule stability (w MiDi metrics) dropped from .87 to .25 per cent). There might be many reasons why this is the case: (i) the best reconstruction loss we found with the equivariant model is higher than the non-equivariant (~9.4e-5 vs. 5.4e-5 MSE on val. set), (ii) the equivariant model needs more capacity to be competitive to the non-equivariant one (currently it has over 90x less parameters), (iii) something on the sampling procedure needs to be different on the equivariant version (unlikely). What makes it even more challenging is that the equivariant network is less efficient (around 50-60% slower) and we faced issues while scaling up to 64^3 grid size (required for GEOM-drugs). We will include a description of what we tried on this space on the appendix.
>
> [D] An end-to-end SE(3)-equivariant segmentation network, Diaz, Geiger, McKinley, 23

---

> > ### Comment · Reviewer_j1qM · 2023-08-18
> > **Response**
> >
> > I thank the authors for the detailed feedback. Though there are still spaces to improve regarding this paper, I retain my positive score since the insights are interesting and the approach yields satisfactory novelty.

---

> > > ### Author Response · Authors · 2023-08-18
> > >
> > > We thank the reviewer for the reviews and feedbacks. We will update the manuscript to make those points more clear. We particularly appreciate that the reviewer agree that a novel/different method with competitive results _is_ a good contribution for this community.

---

### Official Review · Reviewer_R82i · 2023-07-04

**Soundness:** 3 good
**Presentation:** 3 good
**Contribution:** 3 good
**Rating:** 4
**Confidence:** 5

**Summary:**

This paper proposes doing diffusion on voxel space for molecule conformation generation. Because the voxel space is discrete, an efficient sampling method is proposed.


**Strengths:**

This paper proposes doing diffusion on voxel space for molecule conformation generation. Because the voxel space is discrete, an efficient sampling method is proposed.


**Weaknesses:**

## Method:
- It is not clear how the SE(3)-equivariant property is guaranteed in the model. This is critical because once the molecules are too large to fit in the voxel grids, we need to rotate it for a better position.
- The objective function in Eq 2 is not clear. Concretely, it is not clear how the atom types are trained.

## Experiments:
- The authors also referred to MiDi in the paper. I’m wondering why the authors did not add MiDi? Is that because the performance of MiDi is better than VoxMol?
- The MiDi paper introduces more domain-specific metrics, and authors should consider add them.
- Also a closely-related baseline should be cited at least: https://arxiv.org/abs/2305.05708.

## Minor:
- `... capturing long-range dependencies over multiple atoms can become difficult …` It also depends on how the edges are constructured. For point clouds, the edges are often constructured by distance, thus this may not become an issue as long as the spatial distance is close.


**Questions:**

I have some questions on the method section, and would like to confirm with the authors.
- I am wondering what’s the advantage of using NEB for estimating p(x)? Based on lines 135-140, it seems to be a standard EBM with Langevine dynamics, puls an extra NEB module. Is this correct?
- The authors mentioned that `Chains are initialized with Guassian noise with standard deviation`. I would like to double-check what does standard Guassian mean in the voxel grids?

---

> ### Author Rebuttal · Authors · 2023-08-09
>
> We thank the reviewer for the feedback and suggestions. Below we address additional concerns from the reviewer.
>
> **Diffusion model:**
>
> We would like to clarify that the proposed model is _not_ a diffusion model (this is a big point of the paper). Our model is based on neural empirical Bayes and walk-jump sampling (Saremi and Hyvarinen, 19). Although the model is also a generative model based on score functions, it is fundamentally different than diffusion models.
>
> **Advantage of using NEB for estimating p(x):**
>
> In general , it is very difficult to use the score function (the gradient of the loglikelihood of P) to sample from P(x) (space of "clean" molecules). The intuition of NEB is that sampling the noisy molecules y from P(y) is much easier than sampling from highly sparse x in P(x). In NEB, the score function is used to sample noisy molecules y. A single chain of VoxMol sampling can generate multiple noisy molecules (with a fast mixing time in practice). We can then "jump" from noisy to clean molecules with the denoiser network. This is in contrast to diffusion models, where the noise is gradually reduced in the reverse diffusion process (thus one chain per single sample).
>
> **Comparison w. arXiv2305.05708:**
>
> We note this paper appeared 6 days before the submission deadline. We will add this reference to the revised version of the manuscript. What we like about this paper is that—similar to our work—it shows that we can generate molecules with methods that are different from GNN on point clouds. They also show—like we did—that built-in SE(3)-equivariant architecture is not necessary to generate molecules; equivariance can be learned with data/augmentation, as argued in [A, B, C] and related works.
>
> **SE(3) equivariance:**
>
> VoxMol has built-in equivariance to translation and relies on data augmentation through rotation (as it has been done before) to learn rotation equivariance. Unlike arXiv2305.05708, 3D CNNs can in principle be adapted to have built-in rotation equivariance (Weiler et al, 18). Please see SE(3) equivariance paragraph in general rebuttal.
>
> **"…capturing long-range dependencies over multiple atoms can become difficult…":**
>
> We agree that it depends on how the edges are constructed. However, we point that EDM considers interactions between all atoms and uses a fully-connected graph (paragraph after equation 11 on EDM paper). We will clarify that this is a limitation of EDM in particular and not GNNs in general.
>
> **Eq 2 is not clear. Concretely, it is not clear how the atom types are trained:**
>
> Each atom is represented by a different input channel (similar to how R, G and B colors are separated on images). The training is done by denoising every voxel location of every channel of the noisy voxel grids (similar to how image denoisers are applied to every pixel on the three color channels).
>
> **Chain initialization:**
>
> We describe how we initialize the chain on Appendix A3: "We follow [27] and initialize the chains by adding uniform noise to the initial Gaussian noise (with the same σ used during training), i.e., y0 = N(0,σ2Id)+Ud(0,1) (this was observed to mix faster in practice)". y0 is a voxel grid of tensor dimensions `(n_channels, dim, dim, dim)`, where dim is the dimension of the voxel grid (32 on QM9 and 64 on GEOM-drugs) and `n_channel` is the number of atom channels considered. We will move this to the main text for clarity.
>
> [A] Understanding image representations by measuring their equivariance and equivalence, Lenc,Vedaldi, CVPR15
>
> [B] Grounding inductive biases in natural images: invariance stems from variations in data, Bouchacourt,et al,NeurIPS21
>
> [C] The Lie Derivative for Measuring Learned Equivariance. Gruveret al,22

---

> > ### Comment · Reviewer_R82i · 2023-08-14
> > **Follow-ups**
> >
> > Thank you for the replies and correction on the diffusion model. Some minor issues have been addressed. However, my question remains unsolved, and more critical questions come out. Thus, I would like to keep my score.
> >
> > 1. My main concern is on the SE3-equivariance.
> >     - Currently, there is no evaluation metric about the SE3-equivariance in the result tables, especially the rotation. Thus, the `outperforming performance` is debatable.
> >     - You can think of the main baseline, EDM, as a constrained generation model, where the generation process needs to satisfy the SE3-equivariance, while this work is like an unconstrained generation.
> >     - For data augmentation specifically, we can always take rotation as data augmentation. However, as raised in my previous review, what if certain molecules are too large to fit in the voxel grids? How can we guarantee that we can always rotate it to fit in? What about molecules that can never fit it? Will you do truncation? If so, how will this affect the generation? These are the questions critical for data augmentation.
> >     - The related works [A,B,C] conduct experiments on images, which I am unfamiliar with. Is there any work providing similar explorations/observations from the chemistry side?
> >     - Thus, the claim that `data augmentation is sufficient` is not well supported. If you want to prove that generation with data augmentation is sufficient for SE3-equivariance, you need to design a metric specific for group symmetry and evaluate it for all the methods.
> >
> > 2. Thank you for adding MiDi results, and MiDi is an SE3-equivariant method. There are 18 metrics in total, and MiDi performs better than VoxMol on 15 of them.
> >     - As mentioned, MiDi is an SE3-equivariant model, and I'm not sure how authors can do `MiDi+ VoxMol`? Or do you simply mean doing 2D+3D using VoxMol? Then how do you define the covalent bonds in the voxel space? Do they have volumes? If so, then how large? Typically the covalent bonds do not exist in the 3D space, and it's non-trivial to inject them into the voxel space.
> >
> > 3. Yes, as mentioned previously, adding the citation to that paper is sufficient. No need to compare.

---

> > > ### Author Response · Authors · 2023-08-15
> > >
> > > We thank R1 for the reply. We provide additional discussions in the hope of better answering R1's questions (detailed clarifications are posted below).
> > >
> > > 1. **Built-in SE(3)-equivariance (SE3E):**
> > > - Voxel representations have been extensively used in molecular modeling (generative and discriminative). These types of models dont have built-in SE3E but are still useful to the community.
> > > - The point of the paper is not to have a built-in SE3E model, but to show that we achieve competitive empirical results without it (as also shown by arXiv2305.05708).
> > > - Imposing SE3E on top of VoxMol could improve results even further. Future works could include: (i) design 3D CNN architectures with built-in SE3 equivariance, (ii) connect 3D CNNs to SE3E GNNs (as proposed by R2 and in [AA, BB]).
> > > 2. **Comparison with MiDi:**
> > > - MiDi is "2D+3D models" built on the top of a "3D model" (EDM). It shows that 3D models can be augmented with connectivity graphs to improve performance. In principle, this should work for other generative models as well.
> > > - MiDi leverages more information (2D graphs, formal charges) than VoxMol/EDM during training, so quantitative apples-to-apples comparisons are not fair.
> > > - The contributions of this work are (i) proposing a novel model for unconditional generation and (ii) beating SOTA on a challenging dataset, assuming same data is used for training. Leveraging 2D information (on top of 3D atoms) is beyond the scope of this paper.
> > >
> > > We hope we are clear on the clarifications above.
> > >
> > > Do the reviewer agree with authors that (i) there is value in showing that a novel voxel-based model can be competitive with SOTA despite not having built in SE3-equi., and (ii) comparison between Voxmol and MiDi are not apples-to-apples since the models are trained on different data? Please let us know if any other clarification is necessary?
> > >
> > > [AA] Point-Voxel CNN for Efficient 3D Deep Learning, Liu et al, NeurIPS19
> > >
> > > [BB] 3D Shape Generation and Completion through Point-Voxel Diffusion, Zhou et al, ICCV21
> > >
> > > ---
> > > **Detailed clarifications:**
> > > 1. **Built-in SE(3)-equivariance:**
> > > - `outperforming performance`: By "outperforming on GEOM-drugs" we mean that VoxMol has better empirical results than EDM on generative modeling metrics (MiDi metrics, as asked by R1). These numbers are not measuring any "SE3E property", but it clearly shows that the VoxMol better matches the distribution of GEOM-drugs and generates more realistic molecules.
> > > - _What if certain molecules are too large to fit in the voxel grids?_
> > >   - With a fixed grid size, voxel representations are limited to a fixed volume in space (L127). We can only use molecules that would fit the grid for training, and the model can only generate molecules that fit the grid.
> > >   - With a 64^3 voxel grid (at resolution .25A), we can fit over 99% of the drug-like molecules from the GEOM-drugs dataset (L207-209), and train w 1 GPU. Rotating molecules with any rotation does not change that.
> > >   - Every model has limitations. In the case of EDM, the model sampling time scales with n^2, n the number of atoms (since they use fully-connected graphs). Voxels, on the other hand, have constant inference time wrt n (for a fixed grid size) (L257-259). The tradeoff between EDM/VoxMol wrt to voxel grid length and the n of atoms has been discussed in the main text. We will make it clearer on the revised version.
> > > - _Related works [A,B,C]_: this is a very new and active research direction and, as far as we know, they haven't been tested on molecular data yet. This work, in addition to, eg, arXiv2305.05708, shows empirically that there might be something interesting to learn about this on molecular data.
> > > - `data augmentation is sufficient`: We agree with R1 and we did not mention it is sufficient, nor wanted to prove it. We pointed to a few references [A,B,C] that show that equivariance can be learned in neural networks (eg, vision transformers are sota on images w/o having built-in translation or rotation equivariances–they learn it). We agree it is future work to see how data augmentation helps learn rotation equivariance.
> > >
> > > 2. **Comparisons w MiDi:**
> > > - By `MiDi+VoxMol` we meant methods that could leverage 2D information on the top of VoxMol similar to how MiDi leverages 2D information on top of EDM. Eg, the model could use GNN for modeling connectivity graphs (2D) on the top of VoxMol (3D) and train them together/separately. It is an open question on how this can be done efficiently.
> > > - MiDi leverages more information than VoxMol/EDM during training, so quantitative apple-to-apples comparison is not fair.
> > > - Leveraging 2D information (on top of 3D atoms) is beyond the scope of this paper.
> > > - Finally, the results R1 is comparing (MiDi-adapt) appeared _after_ the submission of the conference and are (still) unpublished.

---

> > > > ### Comment · Reviewer_R82i · 2023-08-17
> > > > **Thank You for the Detailed Reply**
> > > >
> > > > Hi authors,
> > > >
> > > > Thank you for the detailed reply, and I am happy to share more insights/concerns.
> > > >
> > > > 1. First, let me explain why I gave a score of 4. In terms of the method, I am not convinced that the generative modeling without SE(3)-equivariance would work (too many ways to define **work**, and I am talking about it from the math/physics). But there are solutions to solve this, as will be explained below, so don't worry. Such a technical concern would lead to a score of 3. Then I noticed the performance was quite good, and I turned the score to 4. The reason I wrote this is to tell the authors that there are still merits in this paper.
> > > > 2. Then, I will go to my main concern. Generative modeling is essentially learning the distribution. This means that SE(3)-equivariant generative modeling is doing the SE(3)-equivariant distribution learning. If there's a metric on the rotation/translation-equivariance, then this metric would be 100% for the SE(3)-equivariant methods, but it is not guaranteed on the voxel method proposed in this work. If the author can give me some numbers to show that modeling on voxel with data augmentations can reach, e.g., `>95% rotation/translation-equivariant  socre` (you need to design this metric yourself), then this can convince me. And I believe that this would be a more interesting and fundamental finding for the whole community.
> > > > 3. The reason why I recommended you to try MiDi is that if your methods can beat MiDi's performance, such kind of `implicit comparison` can be a little much stronger in showing that voxel modeling can work. Yet, I need to point out that such a kind of `implicit comparison` is less persuasive than the one I showed above. Plus now it seems no luck with comparison to MiDi, so you can just skip this experiment.
> > > >
> > > > I hope I have explained my concerns more clearly.

---

> > > > > ### Author Response · Authors · 2023-08-18
> > > > >
> > > > > We thank the review for the engagement and for clarifying their concerns. Below we provide additional discussions in the hope of better answering R1's questions.
> > > > >
> > > > > - **Comparison to MiDi:**
> > > > >   - As mentioned before, `implicit comparison` or any comparison between models trained with different data are not meaningful.
> > > > >   - Independently of this fact, R1 is comparing the performance of our model with a paper that appeared on arxiv on _June 5_ while the submission deadline was _May 15_.
> > > > >   - Does the reviewer agree that benchmarking with a model that appeared _after_ the submission deadline should not influence the ratings of the manuscript?
> > > > >   - If compared with the v1 of that paper (which was on arxiv on May 15), VoxMol outperforms MiDi in most metrics. But again, this does not matter because comparisons are not apples-to-apples. What it tells us is that if we manage to leverage connectivity graphs and formal charges efficiently on top of VoxMol (as MiDi did on top of EDM), we can boost its performance similar to how MiDi boosts EDM.
> > > > > - **" I am not convinced that the generative modeling without SE(3)-equivariance would work (too many ways to define** work **, and I am talking about it from the math/physics)":**
> > > > >   - The machine learning community evaluates deep generative models by comparing the quality of the generated samples with a hold-out set of real samples. This is the definition of  **work** most used by the community and the one we adopt. Better the empirical distribution of generated samples matches the empirical distribution of the hold-out set, then better the generative model.
> > > > >   - Every paper on unconditional molecule generation before this evaluates the quality of their model this way. This includes papers that use voxels, point clouds, surfaces and even text (arXiv2305.05708) as data representation. We do the same.
> > > > >   - Therefore, proposing new metrics to evaluate SE3-equivariant property of generative models is not our objective. We focus on benchmarking generative models in the same way other works do.
> > > > > - There are many published papers on voxel-based generative modeling for molecules (unconditional generation, conditional generation on pockets, scaffolds, fragments, etc). Can these works, or arXiv2305.05708 (mentioned by R1), convince R1 that models without built-in SE(3)-equivariance can generate molecules?
> > > > > - **Metric on the rotation/translation-equivariance:**
> > > > >   - Generative models for molecules output a set of atoms and their coordinates (eg, `xyz` files), independent of the data representation used for training.
> > > > >   - A `rotation/translation-equivariant score` metric makes sense on discriminative tasks, where the _input_ is a molecule (a rotated input molecule should have the same output as non-rotated input). In generative modeling, it is the _output_ that is a molecule while the input is only noise. Since it makes no sense to compute equivariance score on set of atoms, such metric should not be used to measure quality of generative models.
> > > > >
> > > > > Were we clear in (i) how we compare performance of generative models and (ii) the comparisons with MiDi are not apples-to-apples because the training assumptions are different and the model compared appeared after the NeuriPS submission?
> > > > > Please, let us know if any other clarification is necessary.

---

> > > > > > ### Comment · Reviewer_R82i · 2023-08-18
> > > > > > **Quick Reply**
> > > > > >
> > > > > > Hi there,
> > > > > >
> > > > > > Thanks for the reply.
> > > > > >
> > > > > > - For the SE(3)-equivariance, I have explained it is the distribution learning that matters. Not the inference. What you were talking about is the inference (v.s. discriminative).
> > > > > >     - I agree that `proposing new metrics to evaluate SE3-equivariant property of generative models is not our objective`, but you need to make sure your learned distribution satisfies the underlying physics law.
> > > > > >     - Works like EDM, and MiDi satisfy this by design (`this metric would be 100% for the SE(3)-equivariant methods`).
> > > > > >     - But it's not clear on this work. So rigorously, you should prove it.
> > > > > >     - Existing metrics cannot reveal the physical properties in distribution learning.
> > > > > >     - `Every paper on unconditional molecule generation before this evaluates the quality of their model this way.` Yes, works like EDM and MiDi are unconditional, but they are constrained distribution/generation learning. These two (conditional and constrained) are different.
> > > > > > - For MiDi, I have explained that if your metrics cannot beat MiDi, which is fine. `so you can just skip this experiment`
> > > > > > - I'm not sure if arXiv2305.05708 is accepted somewhere. And yes, it has the same issue. I haven't asked you to put too much emphasis on it, just `should be cited at least`.

---

> > > > > > > ### Author Response · Authors · 2023-08-18
> > > > > > >
> > > > > > > We thank the reviewer for the reply. Below we provide additional comments.
> > > > > > > - **SE(3)-equivariance:**
> > > > > > >   - We see your point. What we are trying to point out is that in our paper we are evaluating on the standard way the machine learning community evaluates generative models.
> > > > > > >   - Table 2 shows that, eg, VoxMol has 70.8% molecular stability vs 40.3% of EDM, VoxMol captures much better the distribution of atom types, bond types and the angle between bonds of molecules. These metrics are not perfect, but this huge statistical difference has to be acknowledged. Other SE(3)-equivariant methods before EDM do not even try on GEOM-drugs because its too challenging.
> > > > > > >   - If we understand correctly, the reviewer argues that adding SE(3)-equivariance inductive bias on the NN architecture makes the generated molecules respect the "underlying physical law". This is not true.  A model can have SE(3)-equivariance built-in and still generates "non-physical" molecules.
> > > > > > >   - A trivial example would be to, eg, train EDM with a for just one epoch (or with a learning rate 1000x). The model still has built-in SE(3)-equivariance but will likely output noise (therefore not respecting any "underlying physical law"). In the case of published EDM, 60% of the generated molecules are not stable (meaning that at least one atom has wrong valence). Also, it can't capture the angles between bonds. This clearly means the samples do not always satisfy "underlying physical law".
> > > > > > >   - Built-in SE(3)-equivariance is an inductive bias that allows a neural network to learn more efficiently, especially low-data/discriminative regime. By no means it guarantees that physical laws of generated samples are respected.
> > > > > > >
> > > > > > > - **Comparison with MiDi:** Do we agree that not comparing with MiDi should not be considered a weakness of this submission?
> > > > > > >
> > > > > > > Please, let us know if any other clarification is necessary.

---

> > > > > > > > ### Comment · Reviewer_R82i · 2023-08-18
> > > > > > > > **Reply**
> > > > > > > >
> > > > > > > > My questions remain.
> > > > > > > >
> > > > > > > > 1. You need to either provide your method can reach comparatively good SE(3)-equivariance property and can beat the SOTA performance using SE(3)-equivariance.
> > > > > > > > 2. Or another strategy is, for MiDi, which is 2D+3D, your method is better even though you are not using 2D information. But now the second strategy seems to be a dead-end, yet the weakness remains.
> > > > > > > > 3. I am talking about the SE(3)-equivariant distribution learning in math, not sampling or inference. I think the authors remain confused on this point.
> > > > > > > > 4. I don't know why `EDM can't capture the angles between bonds`? EDM is purely 3D, so where do the 2D covalent bonds come from? And why does EDM not satisfy the physical law? Also, are the authors implying that, because EDM doesn't satisfy physical law, this paper can also ignore the physical law?
> > > > > > > >
> > > > > > > > I am happy to follow up when the authors can provide more rigorous and quantitative support for this work's SE(3)-equivariance property.

---

> > > > > > > > ### Author Response · Authors · 2023-08-18
> > > > > > > > **On "rigorously proving" SE(3) equivariance**
> > > > > > > >
> > > > > > > > We respectfully disagree that we should "rigorously prove" SE3 equivariance. Even a convolutional neural network breaks equivariance due to the presence of non-linearities. We appreciate this discussion, but we have found it very surprising that the reviewer's main reason to give low scores is a philosophical view on SE3 equivariance for a machine learning model. As we pointed out earlier there are very important and impactful models and architectures in machine learning that break equivariance from the start, the most prominent of which are Vision Transformers.

---

### Author Rebuttal · Authors · 2023-08-09

(R IDs: R1=R82i , R2=j1qM, R3=bpDM, R4= zHHZ)

We thank the reviewers for the detailed and helpful reviews. In particular, we thank the suggestion of evaluating on MiDi metrics [R1, R4]. The results on these metrics corroborate the findings of the submission: VoxMol performs slightly worse than EDM on QM9 (Table 1 on the attached pdf), while considerably outperforming EDM on harder GEOM-drugs (Table 2). The metrics shows more clearly that VoxMol (i) beats the current state of art on a (arguably) more realistic/useful dataset _by a large margin_, and (ii) seems to be more scalable. We also acknowledge that reviewers find that this work (iii) is novel in terms of data representation, training and/or sampling [R1, R2, R3, R4], (iv) has good evaluations/ablations [R2, R4], (v) is well written [R2, R3, R4], and (vi) can "shed light on new research directions of connecting 3D CNNs to equivariant GNNs" [R2].

PS: we fixed a bug on the rotation augmentation (randomly rotates the axis between [0, 2] instead of [0,2pi] radians), see `_random_rot_matrix()` function on `./dataset/dataset.py` (submitted source code). The bug fix improves the uniqueness score while other metrics mostly remain the same. Table 1,2 shows results with VoxMol trained with the correct rotation augmentation.

Next, we address the main concerns from reviewers.

**MiDi metrics [R1, R4]:**

First, we run the official MiDi evaluation on EDM's .xyz files (provided by authors) to reproduce results from the MiDi paper. Then, we run the same evaluation on VoxMol .xyz files. Table 1 and 2 on the attached pdf show results for QM9 and GEOM-drugs respectively. To summarize:
* In QM9, VoxMol performs slightly worse than/similar to EDM in most metrics (except on bond angle W1 where the gap is large).
* In GEOM, VoxMol is considerably better (by larger margins in most cases) than EDM in all metrics except on bond length W1, where they perform relatively close.

We will use these tables/metrics on the revised version of the manuscript.

**Comparison w. MiDi [R1]:**

MiDi is built on the top of EDM by leveraging molecular graphs (2D) as well as 3D atoms. Its contributions are therefore orthogonal to ours. In fact, a "VoxMol+MiDi" model could improve results the same way MiDi (3D+2D) improved over EDM (3D). Table1,2 show MiDi results for reference. We note that VoxMol outperforms MiDi (published on ICLR23 MLDD Workshop) on GEOM-drugs in most metrics _without_ using molecular graphs or formal charges information during training.

**SE(3) equivariance [R1, R2, R3]:**

Built-in rotation equivariance is a good property for a network to have, however equivariance can also be learned with strong data augmentation/larger datasets ([A, B, C] and related works). By choosing a 3D U-Net architecture, we lose built-in rotation equivariance but win on expressiveness of the denoising network (U-Nets have been highly optimized for denoising grids, eg, modern image generation). Our experiments show that an efficient denoiser can scale up better, allowing VoxMol to outperform current SOTA on GEOM-drugs despite not having built-in rotation equivariance.

3D CNNs can in theory be adapted to have built-in rotation equivariance (Weiler et al, 18). Successfully designing SE(3)-equivariant 3D CNNs for this problem can potentially improve results further. Currently, equivariant 3D CNNs do not scale as well as the standard ones (and it is a challenge to apply to large datasets). We hope our results motivate the community to explore more scalable SE(3)-equivariant 3D CNNs. We see these developments as architecture improvements and not as the main contributions of the paper.

**Guide/Conditional generation [R3, R4]:**

Like diffusion models (DM), our method also leverages (learned) score functions and relies on Langevin MCMC for sampling. Therefore, in theory we can condition VoxMol similarly to how it is done in diffusion models: by constraining the score function as we walk through the MCMC chain. In the case of DMs, the score function of all T steps are constrained to guide the transition steps from noise to a (conditioned) sample. In VoxMol, the constrained score function would affect the "walk steps" (the Langevin MCMC steps): it would restrict the region where the chain samples noisy molecules y to P(y|c) (instead of P(y)), c is the condition (eg, gradient of a classifier). The "jump step" (a forward pass of the denoising network over the noised molecules) is independent of the condition and remains unchangeable.

Many of the innovations on conditioning DMs come from computer vision, where U-nets are usually used. Since VoxMol has the same architecture (albeit 3D instead of 2D), many of the conditioning techniques/tricks used in images may be more easily transferable. Eg, we could in principle use the gradient of a classifier (trained jointly) to guide the sampling (using the same trick as in Dhariwal and Nichol 21) or adapt gradient-free guidance (Ho and Salisman, 22). Pocket conditioning could also be possible, as in eg, (Schneuing et al. 22, Guan et al. 23). In-painting (related to linker/scaffold/fragments cond.) has also proven to work very well in 2D U-Nets, so it could potentially work with 3D U-Nets as well.

We will add a section in the appendix on how we can condition the generation in different ways. We agree that conditional molecule generation is a more interesting problem in practice. We note that our approach is the first of its kind and the main focus of this submission is to show that (i) it is a feasible approach (this is non-trivial) and (ii) it scales well on unconditional generation, beating current SOTA on a large dataset.

[A] Understanding image representations by measuring their equivariance and equivalence, Lenc,Vedaldi, CVPR15

[B] Grounding inductive biases in natural images: invariance stems from variations in data, Bouchacourt,et al,NeurIPS21

[C] The Lie Derivative for Measuring Learned Equivariance. Gruveret al,22

---

### Decision · Program_Chairs · 2023-09-21

**Decision:**

Accept (poster)

**Comment:**

This paper presents a new approach to molecular generation based on denoising voxel grids. By using a voxel-grid representation, this work takes a vision-based approach to molecular generation instead of graph-based or point clouds methods. The method has strong empirical performance. The paper is well written and includes source code.  While the reviews are mixed, the authors provide extensive discussion during rebuttals that I felt addressed their concerns. Finally, it is unclear how useful the task of unconditional generation is in practice, however, the authors have stated that they will include a discussion of different ways to condition the generation process, which will be a valuable contribution.